# SuperDeepFool: a new fast and accurate minimal adversarial attack

**Alireza Abdollahpoorrostam**
EPFL
Lausanne, Switzerland
alireza.abdollahpoorrostam@epfl.ch

**Mahed Abroshan**
Imperial College, London, UK
m.abroshan23@imperial.ac.uk

**Seyed-Mohsen Moosavi-Dezfooli**
Apple
Zürich, Switzerland
smoosavi@apple.com

## Abstract

Deep neural networks have been known to be vulnerable to adversarial examples, which are inputs that are modified slightly to fool the network into making incorrect predictions. This has led to a significant amount of research on evaluating the robustness of these networks against such perturbations. One particularly important robustness metric is the robustness to minimal $\ell_2$ adversarial perturbations. However, existing methods for evaluating this robustness metric are either computationally expensive or not very accurate. In this paper, we introduce a new family of adversarial attacks that strike a balance between effectiveness and computational efficiency. Our proposed attacks are generalizations of the well-known DeepFool (DF) attack, while they remain simple to understand and implement. We demonstrate that our attacks outperform existing methods in terms of both effectiveness and computational efficiency. Our proposed attacks are also suitable for evaluating the robustness of large models and can be used to perform adversarial training (AT) to achieve state-of-the-art robustness to minimal $\ell_2$ adversarial perturbations.

## 1 Introduction

Deep learning has achieved breakthrough improvement in numerous tasks and has developed as a powerful tool in various applications, including computer vision [32] and speech processing [35]. Despite their success, deep neural networks are known to be vulnerable to adversarial examples, carefully perturbed examples perceptually indistinguishable from original samples [54]. This can lead to a significant disruption of the inference result of deep neural networks. It has important implications for safety and security-critical applications of machine learning models.

Our goal in this paper is to introduce a parameter-free and simple method for accurately and reliably evaluating the adversarial robustness of deep networks in a fast and geometrically-based fashion. Most of the current attack methods rely on general-purpose optimization

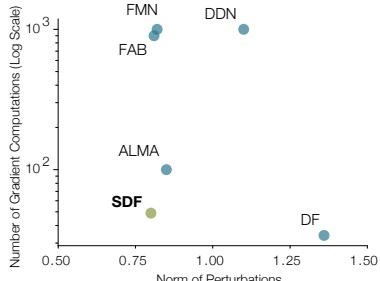

Figure 1: The average number of gradient computations vs the mean $\ell_2$-norm of perturbations. It shows that our novel fast and accurate method, SDF, outperforms other minimum-norm attacks. SDF finds significantly smaller perturbations compared to DF, with only a small increase in computational cost. SDF also outperforms other algorithms in optimality and speed. The numbers are taken from Table 5.

techniques, such as Projected Gradient Descent (PGD) [33] and Augmented Lagrangian [49], which are oblivious to the geometric properties of models. However, deep neural networks' robustness to adversarial perturbations is closely tied to their geometric landscape [14, 29, 38, 41]. Given this, it would be beneficial to exploit such properties when designing and implementing adversarial attacks. This allows to create more effective and computationally efficient attacks on classifiers. Formally, for a given classifier $\hat{k}$ and input $\boldsymbol{x}$, we define an adversarial perturbation as the minimal perturbation $\boldsymbol{r}$ that is sufficient to change the estimated label $\hat{k}(\boldsymbol{x})$:

$$\Delta(\boldsymbol{x}; \hat{k}) := \min_{\boldsymbol{r}} \|\boldsymbol{r}\|_2 \text{ s.t } \hat{k}(\boldsymbol{x} + \boldsymbol{r}) \neq \hat{k}(\boldsymbol{x}). \tag{1}$$

DeepFool (DF) [36] was among the earliest attempts to exploit the "excessive linearity" [23] of deep networks to find minimum-norm adversarial perturbations. However, more sophisticated attacks were later developed that could find smaller perturbations at the expense of significantly greater computation time.

In this paper, we exploit the geometric characteristics of minimum-norm adversarial perturbations to design a family of fast yet simple algorithms that achieves a better trade-off between computational cost and accuracy in finding $\ell_2$ adversarial perturbations (see Fig. 1). Our proposed algorithm, guided by the characteristics of the optimal solution to Eq. (1), enhances DF to obtain smaller perturbations, while maintaining simplicity and computational efficiency that are only slightly inferior to those of DF. Our main contributions are summarized as follows:

- We introduce a novel family of fast yet accurate algorithms to find minimal adversarial perturbations. We conduct a comprehensive evaluation of our algorithms against state-of-the-art (SOTA) adversarial attack methods across multiple scenarios. Our findings demonstrate that our algorithm identifies minimal yet accurate perturbations with significantly greater efficiency than competing SOTA approaches (4).

- Our algorithms are developed in a systematic and well-grounded manner, based on theoretical analysis (3).

- We further improve the robustness of state-of-the-art image classifiers to minimum-norm adversarial attacks via adversarial training on the examples obtained by our algorithms (4.3).

- We significantly improve the time efficiency of the state-of-the-art Auto-Attack (AA) [12] by adding our proposed method to the set of attacks in AA (4.3).

- We revisit the importance of minimal adversarial perturbations as a proxy to demystify deep neural network properties (Appendix G, Appendix O).

**Related works.** It has been observed that deep neural networks are vulnerable to adversarial examples [23, 36, 54]. To exploit this vulnerability, a range of methods have been developed for generating adversarial perturbations for image classifiers. These attacks occur in two settings: white-box, where the attacker has complete knowledge of the model, including its architecture, parameters, defense mechanisms, etc.; and black-box, where the attacker's knowledge is limited, mostly relying on input queries to observe outputs [10, 44]. Further, adversarial attacks can be broadly categorized into two categories: bounded-norm attacks (such as FGSM [23] and PGD [33]) and minimum-norm attacks (such as DF and C&W [6]) with the latter aimed at solving Eq. (1). In this work, we specifically focus on white-box minimum $\ell_2$-norm attacks.

The authors in [54] studied adversarial examples by solving a penalized optimization problem. The optimization approach used in [54] is complex and computationally inefficient; therefore, it cannot scale to large datasets. The method proposed in [23] applied a single-step of the input gradient to generate adversarial examples efficiently. DF was the first method to seek minimum-norm adversarial perturbations, employing an iterative approach. It linearizes the classifier at each step to estimate the minimal adversarial perturbations efficiently. C&W attack [6] transform the optimization problem in [54] into an unconstrained optimization problem. C&W leverages the first-order gradient-based optimizers to minimize a balanced loss between the norm of the perturbation and misclassification confidence. Inspired by the geometric idea of DF, FAB [11] presents an approach to minimize the norm of adversarial perturbations by employing complex projections and approximations while maintaining proximity to the decision boundary. By utilizing gradients to estimate the local geometry of the boundary, this method formulates minimum-norm optimization without the need for tuning a weighting term. DDN [48] uses projections on the $\ell_2$-ball for a given perturbation budget $\epsilon$. FMN [40]

extends the DDN attack to other $\ell_p$-norms. By formulating (1) with Lagrange's method, ALMA [49] introduced a framework for finding adversarial examples for several distances.

**Why does $\ell_2$ white-box adversarial robustness matter?** The reasons for using $\ell_2$ norm perturbations are manifold. We acknowledge that $\ell_2$ threat model may not seem particularly realistic in practical scenarios (at least for images); however, it can be perceived as a basic threat model amenable to both theoretical and empirical analyses, potentially leading insights in tackling adversarial robustness in more complex settings. The fact that, despite considerable advancements in AI/ML, we are yet to solve adversarial vulnerability, motivates part of our community to return to the basics and work towards finding fundamental solutions to this issue [9, 25, 34]. In particular, thanks to their intuitive geometric interpretation, $\ell_2$ perturbations provide valuable insights into the geometry of classifiers. They can serve as an effective tool in the "interpretation/explanation" toolbox to shed light on what/how these models learn. Moreover, it has been demonstrated that [19, 38], $\ell_2$ robustness has several applications beyond security (for more details on the necessity of robustness to $\ell_p$ norms, please refer to Appendix O).

## 2    DeepFool (DF) and Minimal Adversarial Perturbations

In this section, we first discuss the geometric interpretation of the minimum-norm adversarial perturbations, i.e., solutions to the optimization problem in Eq. (1). We then examine DF to demonstrate why it may fail to find the optimal minimum-norm perturbation. Then in the next section, we introduce our proposed method that exploits DF to find smaller perturbations.

Let $f : \mathbb{R}^d \to \mathbb{R}^C$ denote a $C$-class classifier, where $f_k$ represents the classifier's output associated to the $k$th class. Specifically, for a given datapoint $\boldsymbol{x} \in \mathbb{R}^d$, the estimated label is obtained by $\hat{k}(\boldsymbol{x}) = \mathrm{argmax}_k f_k(\boldsymbol{x})$, where $f_k(\boldsymbol{x})$ is the $k^{\text{th}}$ component of $f(\boldsymbol{x})$ that corresponds to the $k^{\text{th}}$ class. Note that the classifier $f$ can be seen as a mapping that partitions the input space $\mathbb{R}^d$ into classification regions, each of which has a constant estimated label (i.e., $\hat{k}(.)$ is constant for each such region). The decision boundary $\mathscr{B}$ is defined as the set of points in $\mathbb{R}^d$ such that $f_i(\boldsymbol{x}) = f_j(\boldsymbol{x}) = \max_k f_k(\boldsymbol{x})$ for some distinct $i$ and $j$. Additive $\ell_2$-norm adversarial perturbations are inherently related to the geometry of the decision boundary. More formally, Let $\boldsymbol{x} \in \mathbb{R}^d$, and $\boldsymbol{r}^*(\boldsymbol{x})$ be the minimal adversarial perturbation defined as the minimizer of Eq. (1). Then:

> ***Properties of minimal adversarial perturbation*** $\rightarrow \boldsymbol{r}^*(\boldsymbol{x})$:
> ① It is orthogonal to the decision boundary of the classifier $\mathscr{B}$.
> ② Its norm, i.e., $\|\boldsymbol{r}^*(\boldsymbol{x})\|_2$ measures the Euclidean distance between $\boldsymbol{x}$ and $\mathscr{B}$, that is $\boldsymbol{x} + \boldsymbol{r}^*$ lies on $\mathscr{B}$.

We aim to investigate whether the perturbations generated by DF satisfy the aforementioned two conditions. Let $\boldsymbol{r}_{\text{DF}}$ denote the perturbation found by DF for a datapoint $\boldsymbol{x}$. We expect $\boldsymbol{x} + \boldsymbol{r}_{\text{DF}}$ to lie on the decision boundary. Hence, if $\boldsymbol{r}$ is the minimal perturbation, for all $0 < \gamma < 1$, we expect the perturbation $\gamma \boldsymbol{r}$ to remain in the same decision region as of $\boldsymbol{x}$ and thus fail to fool the model.

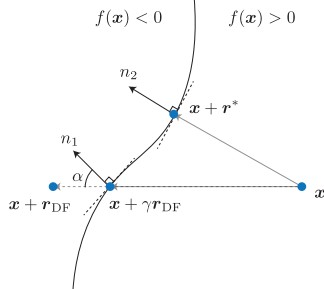

Fig. 2 illustrates the two conditions discussed in Section 2. In the figure, $n_1$ and $n_2$ represent two orthogonal vectors to the decision boundary. The optimal perturbation vector $\boldsymbol{r}^*$ aligns parallel to $n_2$. On the other hand, a non-optimal perturbation $\boldsymbol{r}_{\text{DF}}$ forms an angle $\alpha$ with $n_1$.

In Fig. 3 (left), we consider the fooling rate of $\gamma \, \boldsymbol{r}_{\text{DF}}$ for $0.2 < \gamma < 1$. For a minimum-norm perturbation, we expect an immediate sharp decline for $\gamma$ close to one. However, in Fig. 3 (top-left) we cannot observe such a decline (a sharp decline happens close to $\gamma = 0.9$, not 1). This is a confirmation that DF typically finds an overly perturbed point. One potential reason for this is the fact that DF stops when a misclassified point is found, and this point might be an overly perturbed one within the adversarial region, and not necessarily on the decision boundary.

Figure 2: Illustration of the optimal adversarial example $\boldsymbol{x} + \boldsymbol{r}^*$ for a binary classifier $f$; the example lies on the decision boundary (set of points where $f(\boldsymbol{x}) = 0$) and the perturbation vector $\boldsymbol{r}^*$ is orthogonal to this boundary.

Now, let us consider the other characteristic of the minimal adversarial perturbation. That is, the perturbation should be orthogonal to the decision boundary. We measure the angle between the found

perturbation $r_{\text{DF}}$ and the normal vector orthogonal to the decision boundary ($\nabla f(\boldsymbol{x} + \boldsymbol{r}_{\text{DF}})$). To do so, we first scale $\boldsymbol{r}_{\text{DF}}$ such that $\boldsymbol{x} + \gamma \boldsymbol{r}_{\text{DF}}$ lies on the decision boundary. It can be simply done via performing a line search along $\boldsymbol{r}_{\text{DF}}$. We then compute the cosine of the angle between $\boldsymbol{r}_{\text{DF}}$ and the normal to the decision boundary at $\boldsymbol{x} + \gamma \boldsymbol{r}_{\text{DF}}$ (this angle is denoted by $\cos(\alpha)$). A necessary condition for $\gamma \boldsymbol{r}_{\text{DF}}$ to be an optimal perturbation is that it must be parallel to the normal vector of the decision boundary. In Fig. 3 (right), we show the distribution of cosine of this angle. Ideally, we wanted this distribution to be accumulated around one. However, it clearly shows that this is not the case, which is a confirmation that $\boldsymbol{r}_{\text{DF}}$ is not necessarily the minimal perturbation.

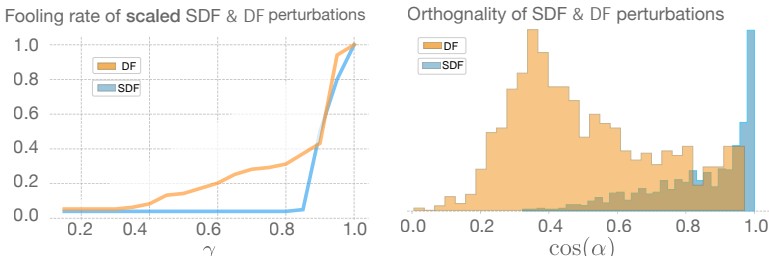

Figure 3: (***Left***) we generated 1000 images with one hundred $\gamma$ between zero and one, and the fooling rate of the DeepFool and SuperDeepFool is reported. This experiment is done on the CIFAR10 dataset and ResNet18 model. (***Right***) histogram of the cosine angle between the normal to the decision boundary and the perturbation vector obtained by DeepFool and SuperDeepFool has been showed.

## 3 SuperDeepFool: Efficient Algorithms to Find Minimal Perturbations

In this section, we propose a new class of methods that modifies DF to address the afore-mentioned challenges in the previous section. The goal is to maintain the desired characteristics of DF, i.e., computational efficiency and the fact that it is parameter-free while finding smaller adversarial perturbations. We achieve this by introducing an additional projection step which its goal is to steer the direction of perturbation towards the optimal solution of Eq. (1).

Let us first briefly recall how DF finds an adversarial perturbations for a classifier $f$. Given the current point $\boldsymbol{x}_i$, DF updates it according to the following equation:

$$\boldsymbol{x}_{i+1} = \boldsymbol{x}_i - \frac{f(\boldsymbol{x}_i)}{\|\nabla f(\boldsymbol{x}_i)\|_2^2} \nabla f(\boldsymbol{x}_i). \qquad (2)$$

Here the gradient is taken w.r.t. the input. The intuition is that, in each iteration, DF finds the minimum perturbation for a linear classifier that approximates the model around $\boldsymbol{x}_i$. The below proposition shows that under certain conditions, repeating this update step eventually converges to a point on the decision boundary.

**Proposition 1** *Let the binary classifier $\mathcal{F}$[1]: $\mathbb{R}^d \to \mathbb{R}$ be continuously differentiable and its gradient $\nabla \mathcal{F}$ is $\beta$-Lipschitz. For a given input sample $\boldsymbol{x}_0$, suppose $\mathcal{B}(\boldsymbol{x}_0, \varepsilon)$ is a ball centered around $\boldsymbol{x}_0$ with radius $\varepsilon$, such that there*

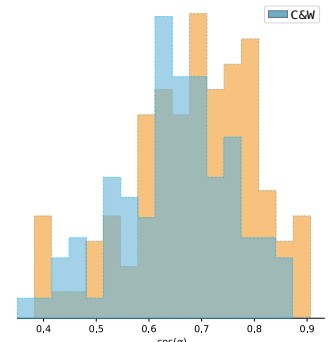

Figure 4: Histogram of the cosine angle between the normal to the decision boundary and the perturbation vector obtained by C&W and FMN.

*exists $\boldsymbol{x}^\star \in \mathcal{B}(\boldsymbol{x}_0, \varepsilon)$ that $f(\boldsymbol{x}^\star) = 0$. If $\|\nabla \mathcal{F}\|_2 \geq \zeta$ for all $\boldsymbol{x} \in \mathcal{B}$ and $\varepsilon < \left(\dfrac{\zeta}{\beta}\right)^2$, then DF iterations converge to a point on the decision boundary.*

*Proof: We defer the proof to the Appendix.*

Notice while the proposition guarantees the perturbed sample to lie on the decision boundary, it does not state anything about the orthogonality of the perturbation to the decision boundary.

To find perturbations that are more aligned with the normal to the decision boundary, we introduce an additional projection step that steers the perturbation direction towards the optimal solution of

---

[1]For the sake of clarity, we use $\mathcal{F}$ to denote binary classifiers for this proposition.

Eq. (1). Formally, the optimal perturbation, $r^*$, and the normal to the decision boundary at $x_0 + r^*$, $\nabla f(x_0 + r^*)$, should be parallel. Equivalently, $r^*$ should be a solution of the following maximization problem:

$$\max_r \frac{r^\top \nabla f(x_0 + r)}{\|\nabla f(x_0 + r)\|\|r\|}, \tag{3}$$

which is the cosine of the angle between $r$ and $\nabla f(x_0 + r)$. A necessary condition for $r^*$ to be a solution of Eq. (3) is that the projection of $r^*$, i.e, $(\mathcal{P}_S)$ on the subspace orthogonal to $\nabla f(x_0 + r^*)$ should be zero. Then, $r^*$ can be seen as a fixed point of the following iterative map:

$$r_{i+1} = T(r_i) = \frac{r_i^\top \nabla f(x_0 + r_i)}{\|\nabla f(x_0 + r_i)\|} \cdot \frac{\nabla f(x_0 + r_i)}{\|\nabla f(x_0 + r_i)\|}. \tag{4}$$

The scalar multiplier on the right-hand side of Eq. (4) represents the norm of the projection of the vector $r_i$ along the gradient direction. The following proposition shows that this iterative process can converge to a solution of Eq. (3).

**Proposition 2** *For a differentiable $f$ and a given $r_0$, $r_i$ in the iterations Eq. (4) either converge to a solution of Eq. (3) or a trivial solution (i.e., $r_i \to 0$).*

*Proof: We defer the proof to the Appendix.*

Intuitively, by the geometrical properties of a decision boundary ($\mathscr{B}$), a small portion of the boundary can be enclosed between two affine parallel hyperplane. The following proposition from ([5]) states that the angle between $\nabla f(x)$ and the optimal direction $\nabla f(x + r^*)$ can be bounded in a neighborhood of the boundary $\mathscr{B}$.

**Proposition 3** *([5]) Given a radius $r > 0$ and $\Psi_r$ is the set of all samples whose distance from the decision boundary $\mathscr{B}$ is less than $r$. For each angle $|\theta| \in \left(0, \frac{\pi}{2}\right)$, there exists a distance $\widetilde{r}_{(\theta)}$, such that, for all $x \in \Psi_{\widetilde{r}_{(\theta)}}$, the following inequality holds:*

$$\frac{\nabla f(x)^\top \nabla f(\mathcal{P}_S(x))}{\|\nabla f(x)\|\|\nabla f(\mathcal{P}_S(x))\|} > \cos(\theta), \tag{5}$$

*where $\mathcal{P}_S$ is the unique projection of $x$ on the $\mathscr{B}$.*

*Proof: We defer the proof to the Appendix.*

### 3.1 A Family of Adversarial Attacks

---
**Algorithm 1:** SDF $(m,n)$ for binary classifiers

---
**Input:** image $x_0$, classifier $f$, $m$, and $n$.
**Output:** perturbation $r$
1 Initialize: $x \leftarrow x_0$
2 **while** $\text{sign}(f(x)) = \text{sign}(f(x_0))$ **do**
3     **repeat** $m$ **times**
4         $x \leftarrow x - \frac{|f(x)|}{\|\nabla f(x)\|_2^2} \nabla f(x)$
5     **end**
6     **repeat** $n$ **times**
7         $x \leftarrow x_0 + \frac{(x-x_0)^\top \nabla f(x)}{\|\nabla f(x)\|^2} \nabla f(x)$
8     **end**
9 **end**
10 **return** $r = x - x_0$

---

Finding minimum-norm adversarial perturbations can be seen as a multi-objective optimization problem, where we want $f(x + r) = 0$ and the perturbation $r$ to be orthogonal to the decision boundary. So far we have seen that DF finds a solution satisfying the former objective and the iterative map Eq. (4) can be used to find a solution for the latter. A natural approach to satisfy both objectives is to **alternate** between these two iterative steps, namely Eq. (2) and Eq. (4). We propose a family of adversarial attack algorithms, coined SuperDeepFool, by varying how frequently we alternate between these two steps. We denote this family of algorithms with SDF$(m, n)$, where $m$ is the number of DF steps Eq. (2) followed by $n$ repetition of the projection step Eq. (4). This process is summarized in Algorithm 1. One interesting case is SDF$(\infty, 1)$ which, in each iteration, continues DF steps till a point on the decision boundary is found and then applies the projection step.

This particular case has a resemblance with the strategy used in [44] to find black-box adversarial perturbations. This algorithm can be interpreted as iteratively approximating the decision boundary with a hyperplane and then analytically calculating the minimal adversarial perturbation for a linear

classifier for which this hyperplane is the decision boundary. It is justified by the observation that the decision boundary of state-of-the-art deep networks has a small mean curvature around data samples [21, 22]. A geometric illustration of this procedure is shown in Figure 5.

## 3.2 SDF Attack

We empirically compare the performance of SDF$(m, n)$ for different values of $m$ and $n$ in Section 4.1. Interestingly, we observe that we get better attack performance when we apply several DF steps followed by a single projection. Since the standard DF typically finds an adversarial example in less than four iterations for state-of-the-art image classifiers, one possibility is to continue DF steps till an adversarial example is found and then apply a single projection step. We simply call this particular version SDF$(\infty, 1)$ of our algorithm SDF, which we will extensively evaluate in Section 4.

SDF can be understood as a generic algorithm that can also work for the multi-class case by simply substituting the first inner loop of Algorithm 1 with the standard multi-class DF algorithm. The label of the obtained adversarial example determines the boundary on which the projection step will be performed. A summary of multi-class SDF is presented in Algorithm 2. Compared to the standard DF, this algorithm has an additional projection step. We will see later that such a simple modification leads to significantly smaller perturbations.

Table 1 demonstrates that SDF family outperforms DF in finding more accurate perturbations, particularly SDF$(\infty,1)$ which significantly outperforms DF at a small cost.

Like any other gradient-based optimization method tackling a non-convex problem, providing a definitive explanation for why one algorithm outperforms others is not straightforward. We have the following speculation on why SDF$(\infty, 1)$ consistently outperforms the other configurations: Note that each projection step reduces the perturbation, while each DF step moves the perturbation nearer to the boundary. So when projection is repeated multiple times ($n > 1$),

Table 1: Comparison of $\ell_2$-norm perturbations using DF and SDF algorithms on CIFAR10, employing consistent model architectures and hyperparameters as those used in [6, 48] studies.

| Attack | Median-$\ell_2$ | Grads |
|---|---|---|
| DF | 0.15 | **14** |
| SDF (1,1) | 0.13 | 22 |
| SDF (1,3) | 0.14 | 26 |
| SDF (3,1) | 0.11 | 30 |
| SDF$(\infty, 1)$ | **0.10** | 32 |

it might undo the progress made by DF, potentially slowing down the algorithm's convergence. On the other hand, by first reaching a boundary point through multiple DF steps and then applying the projection operator just once, we at least ensure that the algorithm has reached intermediate adversarial examples. Each subsequent outer loop is hoped to incrementally move the adversarial example closer to the optimal point (see 5).

---

**Algorithm 2:** SDF for multi-class classifiers

**Input:** image $\boldsymbol{x}_0$, classifier $f$.
**Output:** perturbation $\boldsymbol{r}$
1 Initialize: $\boldsymbol{x} \leftarrow \boldsymbol{x}_0$
2 **while** $\hat{k}(\boldsymbol{x}) = \hat{k}(\boldsymbol{x}_0)$ **do**
3 $\quad \widetilde{\boldsymbol{x}} \leftarrow \texttt{DeepFool}(\boldsymbol{x})$
4 $\quad \boldsymbol{w} \leftarrow \nabla f_{\hat{k}(\widetilde{\boldsymbol{x}})}(\widetilde{\boldsymbol{x}}) - \nabla f_{\hat{k}(\boldsymbol{x}_0)}(\widetilde{\boldsymbol{x}})$
5 $\quad \boldsymbol{x} \leftarrow \boldsymbol{x}_0 + \frac{(\widetilde{\boldsymbol{x}} - \boldsymbol{x}_0)^\top \boldsymbol{w}}{\|\boldsymbol{w}\|^2} \boldsymbol{w}$
6 **end**
7 **return** $\boldsymbol{r} = \boldsymbol{x} - \boldsymbol{x}_0$

---

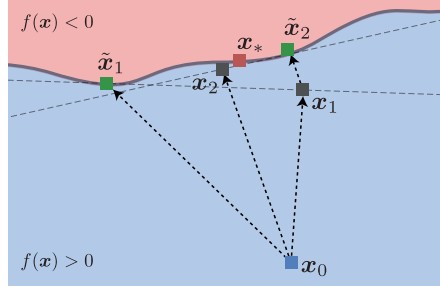

Figure 5: Illustration of two iterations of the SDF$(\infty,1)$ algorithm. Here $\boldsymbol{x}_0$ is the original data point and $\boldsymbol{x}_*$ is the minimum-norm adversarial example.

## 4 Experimental Results

In this section, we conduct extensive experiments to demonstrate the effectiveness of our method in different setups and for several natural and adversarially trained networks. We first introduce our experimental settings, including datasets, models, and attacks. Next, we compare our method with state-of-the-art $\ell_2$-norm adversarial attacks in various settings, demonstrating the superiority of our simple yet fast algorithm for finding accurate adversarial examples. Moreover, we add SDF to the collection of attacks used in AutoAttack, and call the new set of attacks AutoAttack++. This setup

meaningfully speeds up the process of finding norm-bounded adversarial perturbations. We also demonstrate that a model adversarially training using the SDF perturbations becomes more robust compared to the models[2] trained using other minimum-norm attacks. Please refer to Appendix B for details of the experimental setup and metrics.

## 4.1 Comparison with DeepFool (DF)

In this part, we compare our algorithm in terms of orthogonality and size of the $\ell_2$-norm perturbations especially with DF. Assume $r$ is the perturbation vector obtained by an adversarial attack. First, we measure the orthogonality of perturbations by measuring the inner product between $\nabla f(x + r)$ and $r$. As we explained in Section 2, a larger inner product between $r$ and the gradient vector at $f(x + r)$ indicates that the perturbation vector is closer to the optimal perturbation vector $r^*$. We compare the orthogonality of different members of the SDF family and DF.

The results are shown in Table 2. We observe that DF finds perturbations orthogonal to the decision boundary for low-complexity models such as LeNet, but fails to perform effectively when evaluated against more complex ones. In contrast, attacks from the SDF family consistently found perturbations with a larger cosine of the angle for all three models.

Table 2: The cosine similarity between the perturbation vector($r$) and $\nabla f(x + r)$. We performed this experiment on three models trained on CIFAR10.

| Attack | Models | | |
|---|---|---|---|
| | LeNet | RN18 | WRN-28-10 |
| DF | 0.89 | 0.14 | 0.21 |
| SDF (1,1) | 0.90 | 0.63 | 0.64 |
| SDF (1,3) | 0.88 | 0.61 | 0.62 |
| SDF (3,1) | **0.92** | 0.70 | 0.72 |
| SDF ($\infty$, 1) | **0.92** | **0.72** | **0.80** |

**Verifying optimality conditions for SDF.** We validate the optimality conditions of the perturbations generated by SDF using the procedure outlined in Section 2. Comparing Fig. 3 DF and SDF, it becomes evident that our approach effectively mitigates the two issues we previously highlighted for DF. Namely, the alignment of the perturbation with the normal to the decision boundary and the problem of over-perturbation. We can see that unlike DF, the cosine of the angle for SDF is more concentrated around one, which indicates that the SDF perturbations are more aligned with the normal to the decision boundary. Moreover, Fig. 3 shows a sharper decline in the fooling rate (going down quickly to zero) when $\gamma$ decreases. This is consistent with our expectation for an accurate minimal perturbation attack.

## 4.2 Comparison with minimum-norm attacks

We now compare SDF with SOTA minimum $\ell_2$-norm attacks: C&W, FMN, DDN, ALMA, and FAB. For C&W, we use the same hyperparameters as in [48]. We use FMN, FAB, DDN, and ALMA with budgets of 100 and 1000 iterations and report the best performance. For a fair comparison, we clip the pixel-values of SDF-generated adversarial images to $[0, 1]$, consistent with the other minimum-norm attacks. We report the average number of gradient computations per sample, as these operations are computationally intensive and provide a consistent metric unaffected by hardware differences. We also provide a runtime comparison (Appendix Table 19).

We evaluate the robustness of the IBP model, which is adversarially trained on the MNIST dataset, against SOTA

Table 3: We evaluate the performance of iteration-based attacks on MNIST using IBP models, noting the iteration count in parentheses. Our analysis focuses on the best-performing versions, highlighting their significant costs when encountered powerful robust models.

| Attack | FR | Median-$\ell_2$ | Grads |
|---|---|---|---|
| DF | 93.4 | 5.31 | 43 |
| ALMA (1000) | **100** | **1.26** | 1 000 |
| DDN (1000) | 99.27 | 1.46 | 1 000 |
| FAB (1000) | 99.98 | 3.34 | 10 000 |
| FMN (1000) | 89.08 | 1.34 | 1 000 |
| C&W | 4.63 | – | 90 000 |
| SDF | **100** | 1.37 | **52** |

attacks in Table 3. We choose this robust model as it allows us to have a more nuanced comparison between different adversarial attacks. SDF and ALMA are the only attacks that achieve a 100% percent fooling rate against this model, whereas C&W is unsuccessful on most of the data samples. The fooling rates of the remaining attacks also degrade when evaluated with 100 iterations. For instance, FMN's fooling rate decreases from 89% to 67.8% when the number of iterations is reduced from 1000 to 100. This observation shows that, unlike SDF, selecting the *necessary number of iterations* is critical for the success of *fixed-iteration* attacks. Even for ALMA which can achieve

---

[2]We only compare to publicly available models.

a nearly perfect FR, decreasing the number of iterations from 1000 to 100 causes the median norm of perturbations to increase fourfold. In contrast, SDF is able to compute adversarial perturbations using the fewest number of gradient computations while still outperforming the other algorithms, except ALMA, in terms of the perturbation norm. However, it is worth noting that ALMA requires twenty times more gradient computations compared to SDF to achieve a marginal improvement in the perturbation norm.

Table 4 compares SDF with SOTA attacks on the CIFAR10 dataset. The results show that SOTA attacks have a similar norm of perturbations, but an essential point is the speed of attacks. SDF finds more accurate adversarial perturbation very quickly rather than other algorithms.

Table 4: Performance of attacks on the CIFAR-10 dataset with naturally trained WRN-28-10.

| Attacks | FR | Median-$\ell_2$ | Grads |
|---|---|---|---|
| DF | 100 | 0.26 | **14** |
| ALMA | 100 | 0.10 | 100 |
| DDN | 100 | 0.13 | 100 |
| FAB | 100 | 0.11 | 100 |
| FMN | 97.3 | 0.11 | 100 |
| C&W | 100 | 0.12 | 90 000 |
| SDF | 100 | **0.09** | 25 |

We also evaluated all attacks on an adversarially trained model for the CIFAR10 dataset. SDF achieves smaller perturbations with half the gradient calculations than other attacks. SDF finds smaller adversarial perturbations for adversarially trained networks at a significantly lower cost than other attacks, requiring only 20% of FAB's cost and 50% of DDN's and ALMA's (See Tables 11, 19 in the Appendix).

Table 5 demonstrates the performance of SDF on a naturally and adversarially trained models on ImageNet dataset. Unlike models trained on CIFAR10, where the attacks typically result in perturbations with similar norm, the differences between attacks are more nuanced for ImageNet models.

Table 5: Performance comparison of SDF with other SOTA attacks on ImageNet dataset with natural trained RN-50 and adversarially trained RN-50.

| Attack | RN-50 | | | RN-50 (AT) | | |
|---|---|---|---|---|---|---|
| | FR | Median-$\ell_2$ | Grads | FR | Median-$\ell_2$ | Grads |
| DF | 99.1 | 0.31 | **23** | 98.8 | 1.36 | **34** |
| ALMA | **100** | 0.10 | 100 | **100** | 0.85 | 100 |
| DDN | 99.9 | 0.17 | 1,000 | 99.7 | 1.10 | 1,000 |
| FAB | 99.3 | 0.10 | 900 | **100** | 0.81 | 900 |
| FMN | 99.3 | 0.10 | 1,000 | 99.9 | 0.82 | 1,000 |
| C&W | **100** | 0.21 | 82,667 | 99.9 | 1.17 | 52,000 |
| SDF | **100** | **0.09** | 37 | **100** | **0.80** | 49 |

In particular, FAB, DDN, and FMN performance degrades when the dataset changes. In contrast, SDF achieves smaller perturbations at a significantly lower cost than ALMA. This shows that the geometric interpretation of optimal adversarial perturbation, rather than viewing (1) as a non-convex optimization problem, can lead to an efficient solution. On the complexity aspect, the proposed approach is substantially faster than the other methods. In contrast, these approaches involve a costly minimization of a series of objective functions. We empirically observed that SDF converges in less than 5 or 6 iterations to a fooling perturbation; our observations show that SDF consistently achieves SOTA minimum-norm perturbations across different datasets, models, and training strategies, while requiring the least number of gradient computations. This makes it readily suitable to be used as a baseline method to estimate the robustness of very deep neural networks on large datasets.

### 4.3 SDF Adversarial Training (AT)

In this section, we evaluate the performance of a model adversarially trained using SDF against minimum-norm attacks and AutoAttack. Our experiments provide valuable insights into the effectiveness of adversarial training with SDF and sheds light on its potential applications in building more robust models. Adversarial training requires computationally efficient attacks, making costly options such as C&W unsuitable. Therefore, an attack that is parallelizable (both on batch size and gradient computation) is desired for successful adversarial train-

Table 6: The comparison between $\ell_2$ robustness of our adversarial trained model and [48] model.

| Attack | SDF (Ours) | | DDN | |
|---|---|---|---|---|
| | Mean | Median | Mean | Median |
| DDN | 1.09 | 1.02 | 0.86 | 0.73 |
| FAB | 1.12 | 1.03 | 0.92 | 0.75 |
| FMN | 1.48 | 1.43 | 1.47 | 1.43 |
| ALMA | 1.17 | 1.06 | 0.84 | **0.71** |
| SDF | **1.06** | **1.01** | 0.81 | 0.73 |

ing. SDF possesses these crucial properties, making it a promising candidate for building more robust models.

We adversarially train a WRN-28-10 on CIFAR10. Similar to the procedure followed in [48], we restrict $\ell_2$-norms of perturbation to 2.6 and set the maximum number of iterations for SDF to 6.

We train the model on clean examples for the first 200 epochs, and we then fine-tune it with SDF generated adversarial examples for 60 more epochs. Since a model trained using DDN-generated samples [48] has demonstrated greater robustness compared to a model trained using PGD [33], we compare our model with that one (for more details about AT please refer to Appendix O). Our model reaches a test accuracy of 90.8% while the model by [48] obtains 89.0%. SDF adversarially trained model does not overfit to SDF attack because, as Table 6 shows, SDF obtains the smallest perturbation. It is evident that SDF adversarially trained model can significantly improve the robustness of model against minimum-norm attacks up to 30%. In terms of comparison of these two adversarially trained models with AA, our model outperformed the [48] by improving about 8.4% against $\ell_\infty$-AA, for $\varepsilon = 8/255$, and 0.6% against $\ell_2$-AA, for $\varepsilon = 0.5$.

Furthermore, compared to a network trained on DDN samples, our adversarially trained model has a smaller input curvature (Table 7). The second column shows the average spectral-norm of the Hessian w.r.t. input, $\|\nabla^2 f(\mathbf{x})\|_2$, and the third column shows the average of the same quantity normalized by the norm of the input gradient, $\mathcal{C}_f(\mathbf{x}) = \|\nabla^2 f(\mathbf{x})\|_2 / \|\nabla f(\mathbf{x})\|_2$. The standard deviation is denoted by numbers enclosed in brackets.

Table 7: Average input curvature of AT models. According to the measures proposed in [53].

| Model | $\mathbb{E}_{\mathbf{x}}\|\nabla^2 f(\mathbf{x})\|_2$ | $\mathbb{E}_{\mathbf{x}}\mathcal{C}_f(\mathbf{x})$ |
|---|---|---|
| Standard | 600.06 (29.76) | 73.99 (6.62) |
| DDN AT | 2.86 (1.22) | 4.32 (2.91) |
| SDF AT (Ours) | **0.73** (0.08) | **1.66** (0.86) |

This observation corroborates the idea that a more robust network will exhibit a smaller input curvature [1, 37, 39, 42, 47, 53].

**AutoAttack++**

Although it is not the primary focus of this paper, in this section we notably enhance the time efficiency of the AA [12] by incorporating SDF method into the set of attacks in AA.

We introduce a new variant of AA by introducing AutoAttack++ (AA++). AA is a reliable and powerful ensemble attack that contains three types of white-box and a strong black-box attacks. AA evaluates the robustness of a trained model to adversarial perturbations whose $\ell_2/\ell_\infty$-norm is bounded by $\varepsilon$. By substituting SDF with the attacks in the AA, we significantly increase the performance of AA

Table 8: Analysis of robust accuracy for various defense strategies against AA++ and AA with $\varepsilon = 0.5$ for six adversarially trained models on CIFAR10. All models are taken from the RobustBench library [13].

| Models | | AA | | AA++ | |
|---|---|---|---|---|---|
| | Clean acc. | Robust acc. | Grads | Robust acc. | Grads |
| R1 [45] | 95.7% | 82.3% | 1259.2 | **82.1%** | **599.5** |
| R2 [51] | 90.3% | 76.1% | 1469.1 | 76.1% | **667.7** |
| R3 [24] | 89.4% | 63.4% | 1240.4 | **62.2%** | **431.5** |
| R4 [46] | 88.6% | **67.6%** | 933.7 | 68.4% | **715.3** |
| R5 [46] | 89.05% | 66.4% | 846.3 | **62.5%** | **613.7** |
| R6 [15] | 88.02% | 67.6% | 721.4 | **63.4%** | **511.1** |
| Natural | 94.7% | 0.00% | 208.6 | 0.00 | **121.1** |

in terms of ***computational time***. Since SDF is an $\ell_2$-norm attack, we use the $\ell_2$-norm version of AA as well. We restrict maximum iterations of SDF to 10. If the norm of perturbations exceeds $\varepsilon$, we renormalize the perturbation to ensure its norm stays $\leq \varepsilon$. In this context, we have modified the AA algorithm by replacing APGD$^\top$ [12] with SDF due to the former's cost and computation bottleneck in the context of AA (See Appendix F.1 for more details). Our decision to replace APGD$^\top$ with SDF was primarily motivated by the former being a computational bottleneck in AA. As it is shown in Table 8, AA and AA++ achieve similar fooling rates, with AA++ being notably faster. We compared the sets of points that were fooled or not fooled by SDF/APGD$^\top$ across 1000 samples ($\varepsilon = 0.5$). The results indicate that both algorithms fool approximately the same set of points, differing only in a handful of samples for this epsilon value. Therefore, the primary benefit of using SDF is the reduction in computation time. We compare the fooling rate and computational time of AA++ and AA on the models from the RobustBench. In Table 8, we observe that AA++ is up to *three* times faster than AA. In an alternative scenario, we added the SDF to the beginning of the AA set, resulting in a version that is up to two times faster than the original AA, despite now containing five attacks (See Appendix F). This outcome highlights the efficacy of SDF in finding adversarial examples. These experiments suggest that leveraging efficient ***minimum-norm*** and ***non-fixed iteration*** attacks, such as SDF, can enable faster and more reliable evaluation of the robustness of deep models.

## 5 Conclusion and Future Works

In this work, we have introduced a family of parameter-free, fast, and parallelizable algorithms for crafting optimal adversarial perturbations. Our proposed algorithm, SDF, ***consistently*** finds smaller

norm perturbations on various networks and datasets with only a small additional computation cost compared to DF (which is still significantly faster than all SOTA attacks). Furthermore, we have shown that adversarial training using the examples generated by SDF builds more robust models. While our primary focus in this work has been on minimal $\ell_2$ attacks, there exists potential for extending SDF families to other threat models, including general $\ell_p$-norms and targeted attacks. In the Appendix, we have demonstrated straightforward modifications that highlight the applicability of SDF to both targeted and $\ell_\infty$-norm attacks. However, a more comprehensive evaluation remains a direction for future work. Moreover, further limitations of our proposed method are elaborated upon in Appendix N. In the end, by revisiting the necessity of $\ell_p$-norm robustness and characterizing a toy example on robustness-free phenomena, we underscore the pivotal role of minimum-norm attacks in ensuring secure AI systems.

## 6 Acknowledgments

We want to thank Kosar Behnia and Mohammad Azizmalayeri for their helpful feedback. We are very grateful to Fabio Brau and Jérôme Rony for providing code and models and answering questions on their papers.

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

## A Appendix

### A.1 Proofs

**Proof of Proposition 1.**

Since $\nabla \mathcal{F}(\boldsymbol{x})$ is Lipschitz-continuous, for $\boldsymbol{x}, \boldsymbol{y} \in \mathcal{B}(\boldsymbol{x}_0, \varepsilon)$, we have:

$$|\mathcal{F}(\boldsymbol{x}) - \mathcal{F}(\boldsymbol{y}) + \nabla \mathcal{F}(\boldsymbol{y})^T (\boldsymbol{x} - \boldsymbol{y})| \leq \frac{\beta}{2} \|\boldsymbol{x} - \boldsymbol{y}\|^2 \tag{6}$$

DeepFool updates the new $\boldsymbol{x}_n$ in each according to the following equation:

$$\boldsymbol{x}_n = \boldsymbol{x}_{n-1} + \frac{\nabla \mathcal{F}(\boldsymbol{x}_{n-1})}{\|\nabla \mathcal{F}(\boldsymbol{x}_{n-1})\|_2^2} \mathcal{F}(\boldsymbol{x}_{n-1}) \tag{7}$$

Hence if we substitute $\boldsymbol{x} = \boldsymbol{x}_n$ and $\boldsymbol{y} = \boldsymbol{x}_{n-1}$ in (6), we get:

$$|\mathcal{F}(\boldsymbol{x}_n)| \leq \frac{\beta}{2} \|\boldsymbol{x}_n - \boldsymbol{x}_{n-1}\|^2. \tag{8}$$

Now, let $s_n := \|\boldsymbol{x}_n - \boldsymbol{x}_{n-1}\|$. Using (8) and DeepFool's step, we get:

$$s_{n+1} = \frac{\mathcal{F}(\boldsymbol{x}_n)}{\|\nabla \mathcal{F}(\boldsymbol{x}_n)\|} \leq \frac{\beta}{2\zeta} \frac{\mathcal{F}(\boldsymbol{x}_n)^2}{\|\nabla \mathcal{F}(\boldsymbol{x}_n)\|^2} \tag{9}$$

$$s_{n+1} = \frac{\mathcal{F}(\boldsymbol{x}_n)}{\|\nabla \mathcal{F}(\boldsymbol{x}_n)\|} \leqslant s_n \epsilon \frac{\beta^2}{\zeta^2} \tag{10}$$

Using the assumptions of the theorem, we have $\frac{\beta \varepsilon}{\zeta^2} < 1,$ and hence $s_n$ converges to 0 when $n \to \infty$. We conclude that $\{\boldsymbol{x}_n\}$ is a ***Cauchy sequence***. Denote by $\boldsymbol{x}_\infty$ the limit point of $\{\boldsymbol{x}_n\}$. Using the continuity of $\mathcal{F}$ and Eq.(8), we obtain

$$\lim_{n \to \infty} |\mathcal{F}(\boldsymbol{x}_n)| = |\mathcal{F}(\boldsymbol{x}_\infty)| = |\mathcal{F}(\boldsymbol{x}^\star)| = 0, \tag{11}$$

Which concludes the proof of the theorem.

**Proof of Proposition 2.** Let us denote the acute angle between $\nabla f(\boldsymbol{x}_0 + \boldsymbol{r}_i)$ and $\boldsymbol{r}_i$ by $\theta_i$ $(0 \leq \theta_i \leq \pi/2)$. Then from (4) we have $|\boldsymbol{r}_{i+1}| = |\boldsymbol{r}_i| \cos \theta_i$. Therefore, we get

$$|\boldsymbol{r}_{i+1}| = \prod_{i=1}^{i} \cos \theta_i |\boldsymbol{r}_0|. \tag{12}$$

Now there are two cases, either $\theta_i \to 0$ or not. Let us first consider the case where zero is not the limit of $\theta_i$. Then there exists some $\epsilon_0 > 0$ such that for any integer $N$ there exists some $n > N$ for which we have $\theta_n > \epsilon_0$. Now for $\epsilon_0$, we can have a series of integers $n_i$ where for all of them we have $\theta_{n_i} > \epsilon_0$. Since we have $0 \leq |\cos \theta| \leq 1$, we have the following inequality:

$$0 \leq \prod_{i=0}^{\infty} |\cos \theta_i| \leq \prod_{i=0}^{\infty} |\cos \theta_{n_i}| \leq \prod_{i=0}^{\infty} |\cos \epsilon_0| \tag{13}$$

The RHS of the above inequality goes to zero which proves that $\boldsymbol{r}_i \to 0$. This leaves us with the other case where $\theta_i \to 0$. This means that $\cos \theta_i \to 1$ which is the maximum of Eq. (3), this completes the proof.

**Proof of proposition 3 ([5])** We use assumptions discussed in proposition 1 (the continuity of $\nabla f$). We derive that there exists a distance $\boldsymbol{r}$ such that $\|\nabla f(x)\| \neq 0$ in $\bar{\Psi}_{\boldsymbol{r}}$ (the smallest closed set containing $\Psi_{\boldsymbol{r}}$), and so we derive that $\frac{\nabla f}{\|\nabla f\|}$ is uniformly continuous in $\bar{\Psi}_{\boldsymbol{r}}$. Hence, for each $\varepsilon$, there exists a distance $\boldsymbol{r}_\varepsilon \leq \boldsymbol{r}$ such that, for each $\boldsymbol{x}, \boldsymbol{y} \in \bar{\Psi}_{\boldsymbol{r}}$ and $\|\boldsymbol{x} - \boldsymbol{y}\| < \boldsymbol{r}_\varepsilon$, the following inequality holds:

$$\left\| \frac{\nabla f(\boldsymbol{x})}{\|\nabla f(\boldsymbol{x})\|} - \frac{\nabla f(\boldsymbol{y})}{\|\nabla f(\boldsymbol{y})\|} \right\| < \varepsilon, \tag{14}$$

Table 9: Comparison of the effectiveness of line search on the CIFAR10 data for SDF and DF. We use one regularly trained model S (WRN-28-10) and three adversarially trained models (shown with R1 [48], R2 [3] and R3 [43]). ✓ and ✗ indicate the presence and absence of line search respectively.

| Model | DF | | SDF | |
|---|---|---|---|---|
| | ✓ | ✗ | ✓ | ✗ |
| S | 0.16 | 0.19 | **0.09** | 0.10 |
| R1 | 0.87 | 1.02 | **0.73** | 0.76 |
| R2 | 1.40 | 1.73 | **0.91** | 0.93 |
| R3 | 1.13 | 1.36 | **1.04** | 1.09 |

Table 10: Comparison of the effectiveness of line search on the CIFAR-10 data for other attacks. Line search effects are a little for DDN and ALMA. For FMN and FAB because they use line search at the end of their algorithms (they remind this algorithm as a *binary search* and *final search*, respectively), line search does not become effective.

| MODEL | DDN | | ALMA | | FMN | | FAB | |
|---|---|---|---|---|---|---|---|---|
| | ✓ | ✗ | ✓ | ✗ | ✓ | ✗ | ✓ | ✗ |
| WRN-28-10 | 0.12 | 0.13 | 0.10 | 0.10 | 0.11 | 0.11 | 0.11 | 0.11 |
| R1 [48] | 0.73 | 0.73 | 0.71 | 0.71 | 1.10 | 1.10 | 0.75 | 0.75 |
| R2 [3] | 0.96 | 0.97 | 0.93 | 0.94 | 0.95 | 0.95 | 1.03 | 1.03 |
| R3 [43] | 1.04 | 1.04 | 1.06 | 1.06 | 1.08 | 1.08 | 1.07 | 1.07 |

from triangle inequality for norms, we can derive:

$$1 - \frac{1}{2}\varepsilon^2 < \frac{\nabla f(\boldsymbol{x})^T \nabla f(\boldsymbol{y})}{\|\nabla f(\boldsymbol{x})\|\|\nabla f(\boldsymbol{y})\|}. \tag{15}$$

In conclusion, by taking $\boldsymbol{y} = \mathcal{P}_{\mathcal{S}}(\boldsymbol{x})$ and by choosing $\varepsilon = \sqrt{2 - 2\cos(\theta)}$, we achieve upper bound for $\cos(\theta)$ where $\widetilde{\boldsymbol{r}}_{(\theta)} = \min(\boldsymbol{r}_{\max}, \boldsymbol{r}_{\varepsilon})$. Where $\boldsymbol{r}_{\max}$ is a maximum distance such that for each $\boldsymbol{x}$ in the $\Psi_{\boldsymbol{r}_{\max}}$ there exists a $\mathcal{P}_{\mathcal{S}}(\boldsymbol{x}) \in \mathscr{B}$ solves the minimum-norm optimization problem.

# B    Setup

We test our algorithms on architectures trained on MNIST, CIFAR10, and ImageNet datasets. For MNIST, we use a robust model called IBP from [60] and naturally trained model called SmallCNN. For CIFAR10, we use three models: an adversarially trained PreActResNet-18 [27] from [43], a regularly trained Wide ResNet 28-10 (WRN-28-10) from [58] and LeNet [31]. These models are obtainable via the RobustBench library [13]. On ImageNet, we test the attacks on two ResNet-50 (RN-50) models: one regularly trained and one $\ell_2$ adversarially trained, obtainable through the robustness library [18]. We additionally evaluate the robustness of Vision Transformers (ViT-B-16 [17]) and reevaluate the comparative analysis between ViTs and CNNs.

# C    On the benefits of line search

As we show in Figure 3, DF typically finds an overly perturbed point. SDF's gradients depend on DF, so overly perturbing DF is problematic. Line search is a mechanism that we add to the end of our algorithms to tackle this problem. For a fair comparison between adversarial attacks, we add this algorithm to the end of other algorithms to investigate the effectiveness of line search. As shown in Table 9, we observe that line search can increase the performance of the DF significantly. However, this effectiveness for SDF is a little. We now measure the effectiveness of line search for other attacks. As observed from Table 10, line search effectiveness for DDN and ALMA is small.

# D  Comparison on CIFAR10 with the AT PRN-18

In this section, we compare SDF with other minimum-norm attacks against an adversarially trained network [43]. In Table 11, SDF achieves smaller perturbation compared to other attacks, whereas it costs only half as much as other attacks.

Table 11: Comparison of SDF with other state-of-the-art attacks for median $\ell_2$ on CIFAR-10 dataset for adversarially trained network (PRN-18 [43]).

| ATTACK | FR | MEDIAN-$\ell_2$ | GRADS |
|---|---|---|---|
| ALMA | 100 | 0.68 | 100 |
| DDN | 100 | 0.73 | 100 |
| FAB | 100 | 0.77 | 210 |
| FMN | 99.7 | 0.81 | 100 |
| SDF | 100 | **0.65** | **46** |

# E  Performance comparison of adversarially trained models versus Auto-Attack (AA)

Evaluating the adversarially trained models with attacks used in the training process is not a standard evaluation in the robustness literature. For this reason, we evaluate robust models with AA. We perform this experiment with two modes; first, we measure the robustness of models with $\ell_\infty$ norm, and in a second mode, we evaluate them in terms of $\ell_2$ norm. Tables 12 and 13 show that adversarial training with SDF samples is more robust against reliable AA than the model trained on DDN samples [48].

Table 12: Robustness results of adversarially trained models on CIFAR-10 with $\ell_\infty$-AA. We perform this experiment on 1000 samples for each $\varepsilon$.

| MODEL | NATURAL | $\varepsilon = \frac{6}{255}$ | $\frac{8}{255}$ | $\frac{10}{255}$ |
|---|---|---|---|---|
| DDN | 89.1 | 45 | 29.6 | 17.6 |
| SDF (OURS) | 90.8 | **47.5** | **38.1** | **25.4** |

Table 13: Robustness results of adversarially trained models on CIFAR-10 with $\ell_2$-AA. We perform this experiment on 1000 samples for each $\varepsilon$.

| MODEL | NATURAL | $\varepsilon = 0.3$ | 0.4 | 0.5 | 0.6 |
|---|---|---|---|---|---|
| DDN | 89.1 | 78.1 | 73 | 67.5 | 61.7 |
| SDF (OURS) | 90.8 | **83.1** | **79.7** | **68.1** | **63.9** |

# F  Another variants of AA++

As we mentioned, in an alternative scenario, we added the SDF to the beginning of the AA set, resulting in a version that is up to two times faster than the original AA. In this scenario, we do not exchange the SDF with APGD. We add SDF to the AA configuration. So in this configuration, AA has five attacks (SDF, APGD, APGD$^\top$, FAB, Square). By this design, we guarantee the performance of AA. An interesting phenomenon observed from these tables is that when the budget increases, the speed of the AA++ increases. We should note that we restrict the number of iterations for SDF to 10.

## F.1  Why do we replace SDF with APGD$^\top$?

It is well established that AutoAttack (AA) is a robust method for evaluating model robustness, unaffected by gradient obfuscation [2]. The primary limitation of AA, however, is its computational intensity. To thoroughly evaluate a model, it must be subjected to four distinct attacks sequentially.

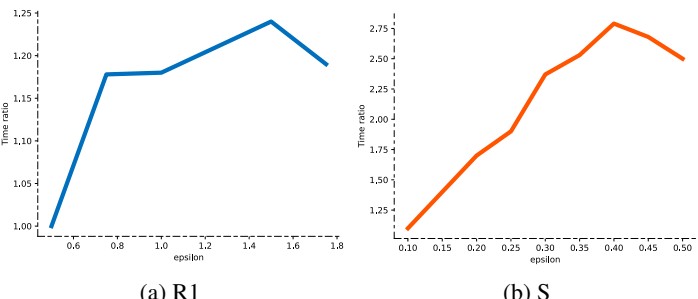

(a) R1                                    (b) S

Figure 6: In this figure, we show the time ratio of AA to AA++. For regularly trained model (WRN-28-10) and adversarially trained model [43] (R1). We perform this experiment on 1000 samples from CIFAR10 data.

Our empirical analysis identified the $APGD^\top$ attack as the main computational bottleneck in AA. For example, when attacking a standard WRN-28-10 model trained on CIFAR-10, $APGD^\top$ requires approximately **4310** backward passes to achieve a $100\%$ fooling rate. Similarly, for an adversarially trained WRN-28-10 [8] model on CIFAR-10, $APGD^\top$ necessitates around **5660** backward passes to attain a $100\%$ fooling rate. To address this issue, rather than simply replacing SDF with another minimum-norm attack such as FAB in AA, we mitigate the bottleneck by employing a faster minimum-norm attack like SDF.

## G    Why do we need stronger minimum-norm attacks?

Bounded-norm attacks like FGSM [23], PGD [33], and momentum variants of PGD [56], by optimizing the difference between the logits of the true class and the best non-true class, try to find an adversarial region with maximum confidence within a given, fixed perturbation size. Bounded-norm attacks only evaluate the robustness of deep neural networks; this means that they report a single scalar value as robust accuracy for a fixed budget. The superiority of minimum-norm attacks is to report a distribution of perturbation norms, and they do not report a percentage of fooling rates (robust accuracy) by a single scalar value. This critical property of minimum-norm attacks helps to accelerate to take an in-depth intuition about the geometrical behavior of deep neural networks.

We aim to address a phenomenon we observe by using the superiority of minimum-norm attacks. We observed that a minor change within the design of deep neural networks affects the performance of adversarial attacks. To show the superiority of minimum-norm attacks, we show how minimum-norm attacks verify these minor changes rather than bounded-norm attacks.

Modeling with max-pooling was a fundamental aspect of convolutional neural networks when they were first introduced as the best image classifiers. Some state-of-the-art classifiers such as [26, 30, 52] use this layer in network configuration. We use the pooling layers to show that using the max-pooling and Lp-pooling layer in the network design leads to finding perturbation with a bigger $\ell_2$-norm.

Assume that we have a classifier $f$. We train $f$ in two modes until the training loss converges. In the first mode, $f$ is trained in the presence of the pooling layer in its configuration, and in the second mode, $f$ does not have a pooling layer. When we measure the robustness of these two networks with regular budgets used in bounded-norms attacks like PGD ($\varepsilon = 8/255$), we observe that the robust accuracy is equal to $0\%$. This is precisely where bounded-norm attacks such as PGD mislead robustness literature in its assumptions regarding deep neural network properties. However, a solution to solve the problem of bounded-norm attack scan be proposed: *"Analyzing the quantity of changes in robust accuracy across different epsilons reveal these minor changes."* Is this case, the solution is costly. This is precisely where the distributive view of perturbations from worst-case to best-case of minimum-norm attacks detects this minor change.

To show these changes, we trained ResNet-18 and Mobile-Net [28] in two settings. In the first setting, we trained them in the presence of a pooling layer until the training loss converged, and in the second setting, we trained them in the absence of a pooling layer until the training loss converged. We should note that we remove all pooling-layers in these two settings. For a fair comparison, we train models until they achieve zero training loss using a multi-step learning rate. We use max-pooling and Lp-pooling, for $p = 2$, for this minor changes.

Table 14: This table shows the $\ell_2$-median for the minimum-norm attacks. For all networks, we set learning rate = 0.01 and weight decay = 0.01. For training with Lp-pooling, we set $p = 2$ for all settings.

| ATTACK | RN18 | | | MOBILENET | | |
|---|---|---|---|---|---|---|
| | NO POOL | MAX-POOL | LP-POOL | NO POOL | MAX-POOL | LP-POOL |
| DF | 0.40 | 0.90 | 0.91 | 0.51 | 0.95 | 0.93 |
| DDN | 0.16 | 0.25 | 0.26 | 0.22 | 0.27 | 0.26 |
| FMN | 0.18 | 0.27 | 0.30 | 0.24 | 0.30 | 0.29 |
| C&W | 0.18 | 0.25 | 0.27 | 0.22 | 0.26 | 0.24 |
| ALMA | 0.19 | 0.23 | 0.23 | **0.20** | 0.25 | 0.22 |
| SDF | **0.16** | **0.21** | **0.22** | **0.20** | **0.23** | **0.21** |

Table 15: This table shows the robust accuracy for all networks against to the AA and PGD. For training with Lp-pooling, we set $p = 2$ for all settings.

| ATTACK | RN18 | | | MOBILENET | | |
|---|---|---|---|---|---|---|
| | NO POOL | MAX-POOL | LP-POOL | NO POOL | MAX-POOL | LP-POOL |
| AA | 1.1% | 17.2% | 16.3% | 8.7% | 21.3% | 20.2% |
| PGD | 9.3% | 28% | 26.2% | 16.8% | 31.4% | 28.7% |

Table 14 shows that using a pooling layer in network configuration can increase robustness. DF has an entirely different behavior according to the presence or absence of the pooling layer; max-pooling affects up to $50\%$ of DF performance. This effect is up to $9\%$ for DDN and FMN. ALMA and SDF show a $4\%$ impact in their performance, which shows their consistency compared to other attacks.

As shown in Table 15, we observe that models with pooling-layers have more robust accuracy when facing adversarial attacks such as AA and PGD. It should be noted that using regular epsilon for AA and PGD will not demonstrate these modifications. For this reason, we choose an epsilon for AA and PGD lower ($\varepsilon = 2/255$) than the regular format ($\varepsilon = 8/255$).

Table 14 and 15 demonstrate that pooling-layers can affect adversarial robustness of deep networks. Powerful attacks such as SDF and ALMA show high consistency in these setups, highlighting the need for powerful attacks.

### G.1 Max-pooling's effect on the decision boundary's curvature

Here, we take a step further and investigate why max-pooling impacts the robustness of models. In order to perform this analysis, we analyze gradient norms, Hessian norms, and the model's curvature. The curvature of a point is a mathematical quantity that indicates the degree of non-linearity. It has been observed that robust models are characterized by their small curvature [37], implying smaller Hessian norms. In order to investigate robustness independent of non-linearity, [53] propose *normalized curvature*, which normalizes the Hessian norm at a given input $x$ by its corresponding gradient norm. They defined *normalized curvature* for a neural network classifier $f$ as $\mathcal{C}_f(x) = \|\nabla^2 f(x)\|_2/(\|\nabla f(x)\|_2 + \varepsilon)$. Where $\|\nabla f(x)\|_2$ and $\|\nabla^2 f(x)\|_2$ are the $\ell_2$-norm of the gradient and the spectral norm of the Hessian, respectively, where $\nabla f(x) \in \mathbb{R}^d$, $\nabla^2 f(x) \in \mathbb{R}^{d \times d}$, and $\varepsilon > 0$ is a small constant to ensure the proper behavior of the measure. In Table 16, we measure these quantities for two trained models, one with max-pooling and one without. It clearly shows that the model incorporating max-pooling exhibits a smaller curvature. This finding corroborates the observation that models with greater robustness tend to have a smaller curvature value.

Table 16: Model geometry of different ResNet-18 models. W (with pooling) and W/O (without pooling).

| MODEL | $\mathbb{E}_x\|\nabla f(x)\|_2$ | $\mathbb{E}_x\|\nabla^2 f(x)\|_2$ | $\mathbb{E}_x\mathcal{C}_f(x)$ |
|---|---|---|---|
| W | **4.75** $\pm\,1.54$ | **120.70** $\pm\,48.74$ | **14.94** $\pm\,0.52$ |
| W/O | 7.04 $\pm\,2.44$ | 269.74 $\pm\,10.23$ | 22.81 $\pm\,2.58$ |

Table 17: Model geometry for regular and adversarially trained models.

| MODEL | $\mathbb{E}_{\boldsymbol{x}}\|\nabla f(\boldsymbol{x})\|_2$ | $\mathbb{E}_{\boldsymbol{x}}\|\nabla^2 f(\boldsymbol{x})\|_2$ | $\mathbb{E}_{\boldsymbol{x}}\mathcal{C}_f(\boldsymbol{x})$ |
|---|---|---|---|
| STANDARD | $9.54 \pm_{1.02}$ | $600.06 \pm_{29.76}$ | $73.99 \pm_{6.62}$ |
| DDN AT | $0.91 \pm_{0.34}$ | $2.86 \pm_{1.22}$ | $4.32 \pm_{2.91}$ |
| SDF AT | $\mathbf{0.38} \pm_{0.60}$ | $\mathbf{0.73} \pm_{0.08}$ | $\mathbf{1.66} \pm_{0.86}$ |

## H  Model geometry for AT models

In this section we provide curvature analysis of our adversarially trained networks, SDF AT, and DDN AT model. Table 17 shows that our AT model decreases the curvature of network more than DDN AT model.

## I  CNN architecture used in Table 1

| Layer Type | CIFAR-10 |
|---|---|
| Convolution + ReLU | $3 \times 3 \times 64$ |
| Convolution + ReLU | $3 \times 3 \times 64$ |
| max-pooling | $2 \times 2$ |
| Convolution + ReLU | $3 \times 3 \times 128$ |
| Convolution + ReLU | $3 \times 3 \times 128$ |
| max-pooling | $2 \times 2$ |
| Fully Connected + ReLU | 256 |
| Fully Connected + ReLU | 256 |
| Fully Connected + Softmax | 10 |

The architecture used to compare SDF variants and DF (Table 1) is summarized in above Table.

## J    ViT-B-16 for CIFAR-10

Given our available computational resources, we conduct experiments on a ViT-B-16 [17] trained on CIFAR-10, achieving 98.55% accuracy. The results are summarized in the following table:

| Attack | FR (%) | Median-$\ell_2$ | Grads |
|--------|--------|-----------------|-------|
| DF     | 98.2   | 0.29            | 19    |
| ALMA   | 100    | 0.12            | 100   |
| DDN    | 100    | 0.14            | 100   |
| FAB    | 100    | 0.14            | 100   |
| FMN    | 99.1   | 0.15            | 100   |
| C&W    | 100    | 0.15            | 91,208 |
| SDF    | 100    | 0.10            | 32    |

As seen, this transformer model does not exhibit significantly greater robustness compared to CNNs, with only a negligible difference of 0.01 compared to a WRN-28-10 trained on CIFAR-10. These results support the notion that there might not be a substantial disparity between the adversarial robustness of ViTs and CNNs. This aligns with the findings of [4]. They argue that earlier claims of transformers being more robust than CNNs stems from an unfair comparison and evaluation methods. We believe that thorough evaluations using minimum norm attacks could be helpful in resolving this debate.

## K    Natural (Regular) Trained MNIST Model

In Table 18 we show the results of evaluating adversarial attacks on naturally trained SmallCNN on MNIST dataset. Our algorithm demonstrates a higher rate of convergence compared to other algorithms, as the perturbations for all algorithms are generally similar.

Table 18: We compare the performance of all algorithms on the natural SmallCNN model that was trained on the MNIST dataset.

| Attacks | FR | Median-$\ell_2$ | Grads |
|---------|-----|-----------------|-------|
| ALMA    | 100 | **1.34**        | 1000  |
| DDN     | 100 | 1.36            | 1000  |
| FAB     | 100 | 1.36            | 10000 |
| FMN     | 97.10 | 1.37          | 1000  |
| C&W     | 99.80 | 1.35          | 90000 |
| SDF     | 100 | **1.34**        | **67** |

## L    Runtime Comparison

We report the number of gradient computations as a main proxy for computional cost comparison. In Table 19, we have compared the runtime of different attacks for a fixed hardware. SDF is significantly faster.

Table 19: Runtime comparison for adversarial attacks on WRN-28-10 architecture trained on CIFAR10, for both naturally trained model and adversarially trained models.

| Attacks | Natural | | R1 [45] | |
|---------|----------|-----------------|----------|-----------------|
|         | Time (S) | Median-$\ell_2$ | Time (S) | Median-$\ell_2$ |
| ALMA    | 1.71     | 0.10            | 13.10    | 1.22            |
| DDN     | 1.54     | 0.13            | 12.44    | 1.53            |
| FAB     | 2.33     | 0.11            | 16.21    | 1.66            |
| FMN     | 1.42     | 0.11            | 10.25    | 1.83            |
| C&W     | 734.8    | 0.12            | 5402.1   | 1.68            |
| SDF     | **0.48** | **0.09**        | **2.93** | **1.19**        |

# M Query-Distortion Curves

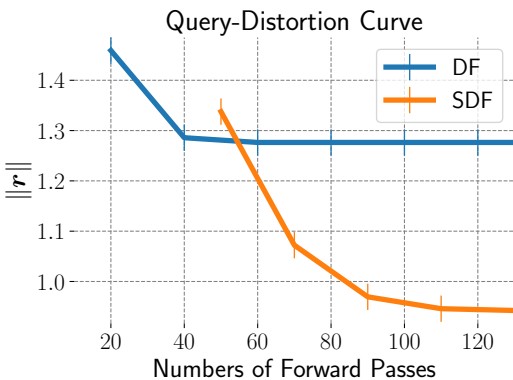

Figure 7: As demonstrated in [40], query-distortion curves are utilised as a metric for evaluating computational complexity of white-box attacks. In this particular context, the term "query" refers to the quantity of forward passes available to find adversarial perturbations.

Unlike FMN and ALMA, SDF (and DF) does not allow control over the number of forward and backward computations. They typically stop once a successful adversarial example is found. Terminating the process prematurely could prevent them from finding an adversarial example. Hence, we instead opted to plot the median norm of achievable perturbations for a given maximum number of queries (Figure 7) Although this is not directly comparable to the query-distortion curves in [40], it provides a more comprehensive view of the query distribution than the median alone.

# N Limitations

In this section, we discuss some limitations and potential extensions of SDF.

**Extension to other $\ell_p$-norms and targeted attacks.** The proposed attack is primarily designed for $\ell_2$-norm adversarial perturbations. Moreover, our method, similar to DeepFool (DF), is non-targeted. Though there are potential approaches for adapting SDF to targeted and $\ell_p$ attacks, these aspects remain largely unexplored in our work.

Nevertheless, we here demonstrate how one could possibly extend SDF to other $p$-norms. A simple way is to replace the $\ell_2$ projection (Line 5 of Algorithm 2) with a projection operator minimizing $\ell_p$ norm similar to the derivations used in [36]. In particular, for $p = \infty$, the following projection would replace the line 5 of Algorithm 2:

$$\boldsymbol{x} \leftarrow \boldsymbol{x}_0 + \frac{(\widetilde{\boldsymbol{x}} - \boldsymbol{x}_0)^\top \boldsymbol{w}}{||\boldsymbol{w}||_1} \text{sign}(\boldsymbol{w}) \tag{16}$$

In Table 20, we compare the performance of this modified version of SDF, named $\text{SDF}_{\ell_\infty}$ with FMN, FAB, and DF, on two pretrained networks M1 [33] and M2 [48] on CIFAR-10 dataset. Our findings indicate that $\text{SDF}_{\ell_\infty}$ also exhibits superior performance compared to other algorithms in discovering smaller perturbations.

Table 20: Performance of $\text{SDF}_{\ell_\infty}$ on two robust networks trained on CIFAR-10 dataset.

| ATTACKS | M1 | | | M2 | | |
|---|---|---|---|---|---|---|
| | MEDIAN $\ell_\infty$ | FR | GRADS | MEDIAN $\ell_\infty$ | FR | GRADS |
| DF | 0.031 | 96.7 | **24** | 0.043 | 97.4 | **31** |
| FAB | 0.025 | 99.1 | 100 | 0.038 | 99.6 | 100 |
| FMN | 0.024 | **100** | 100 | 0.035 | **100** | 100 |
| $\text{SDF}_{\ell_\infty}$ | **0.019** | **100** | 33 | **0.027** | **100** | 46 |

Table 21: Performance of targeted SDF on a standard trained WRN-28-10 on CIFAR-10, measured using 1000 random samples.

| ATTACKS | TARGETED | | | | UNTARGETED | | | |
|---|---|---|---|---|---|---|---|---|
| | FR | MEAN $\ell_2$ | MEDIAN $\ell_2$ | GRADS | FR | MEAN $\ell_2$ | MEDIAN $\ell_2$ | GRADS |
| DDN | 100 | 0.24 | 0.25 | 100 | 100 | 0.13 | 0.14 | 100 |
| FMN | 96.2 | 0.22 | 0.24 | 100 | 97.3 | 0.11 | 0.13 | 100 |
| SDF (TARGETED) | 98.2 | 0.21 | 0.22 | 25 | 100 | 0.10 | 0.11 | 34 |

Furthermore, we can convert SDF to a targeted attack by replacing the line 3 of Algorithm 2 with the targeted version of DeepFool, and the line 4 with the following:

$$\boldsymbol{w} \leftarrow \nabla f_t(\widetilde{\boldsymbol{x}}) - \nabla f_{\hat{k}(\boldsymbol{x}_0)}(\widetilde{\boldsymbol{x}}), \tag{17}$$

where $t$ is the target label. We followed the procedure outlined in [6] to measure the performance in the targeted setting. The result is summarized in Table 21. While SDF is effective in quickly finding smaller perturbations, it does not achieve a 100% fooling rate. Further analysis is required to understand the factors preventing SDF from converging in certain cases. This aspect remains an area for future work.

**Convergence guarantees.** A common challenge for all gradient-based optimization methods applied to non-convex problems is the lack of a guarantee in finding globally optimal perturbations for SotA neural networks. Obtaining even local guarantees is not trivial. Nevertheless, in Propositions 1 and 2 we worked towards this goal. We have established local guarantees showing the convergence of each individual operation, namely the DeepFool step and projection step. However, further analysis is needed to establish local guarantees for the overall algorithm.

**Adaptive attacks.** It is known that gradient-based attacks, ours included, are prone to gradient obfuscation/masking [7]. To counter this challenge, adaptation, as outlined in [55], is needed. It is also important to recognize that adapting geometric attacks such as SDF, does not follow a one-size-fits-all approach, as opposed to loss-based ones such as PGD. While this might be perceived as a weakness, it actually underscores a broader trend in the community. The predominant focus has been on loss-based attacks. This emphasis has inadvertently led to less exploration and development in the realm of geometric attacks.

## O   Vanila Adversarial Training

**Vanila Adversarial Training without Additional Regularization.** Our primary objective was to evaluate which adversarial attacks technique most effectively enhances robustness among PGD [33], DDN [48], and SDF. This focus differs from comparing various adversarial training strategies such as TRADES [59], TRADES-AWP [57], HAT [43], and UIAT [16]. These strategies often include additional regularization techniques to enhance Madry's method using PGD adversarial examples. Therefore, our assertion is not aimed at developing a state-of-the-art robust model. Instead, we aim to demonstrate that vanilla AT, when combined with minimum-norm attacks like SDF, can potentially outperform PGD-based models. Accordingly, we selected vanilla adversarial training with SDF-generated samples for our study and compared its effectiveness against a network trained with DDN samples. While TRADES or similar AT strategies could also integrate SDF, exploring this combination will be addressed in future research endeavors.

**Why $\ell_p$ norm is Critical?** The existing literature has explored a variety of approaches to understanding adversarial examples. For example, training on $\ell_p$-norm adversarial examples has been identified as a form of spectral regularization [50], and adversarial perturbations, seen as counterfactual explanations, have been connected to saliency maps in image classifiers [20]. The rapid and accurate generation of these perturbations is critical for the empirical investigation of such phenomena. Moreover, minimal $\ell_p$ adversarial perturbations are often considered "first order approximations of the decision boundary," illuminating the local geometric characteristics of models near data samples. This insight underscores the need for quick and precise methods for such explorations. Additionally, these

minimal perturbations provide a data-dependent, worst-case analysis of certain test-time corruptions, facilitating worst-case evaluations not only in the input space but also in the transformation space [29]. Within the context of Large Language Models (LLMs), these perturbations could potentially act as probing tools within their embedding space to examine their geometric properties. However, it is important to note that our interest in these topics was driven more by academic curiosity than by their practical applications in this specific study.

# P  Multi-class algorithms for SDF (1,3) and SDF (1,1)

Algorithm (3,4) summarizes pseudo-codes for the multi-class versions of $SDF(1, 1)$ and $SDF(1, 3)$.

---

**Algorithm 3:** SDF (1,1)

**Input:** image $\boldsymbol{x}$, classifier $f$.
**Output:** perturbation $\boldsymbol{r}$

1  Initialize: $\boldsymbol{x}_0 \leftarrow \boldsymbol{x}, \ i \leftarrow 0$
2  **while** $\hat{k}(\boldsymbol{x}_i) = \hat{k}(\boldsymbol{x}_0)$ **do**
3    **for** $k \neq \hat{k}(\boldsymbol{x}_0)$ **do**
4      $\boldsymbol{w}_k' \leftarrow \nabla f_k(\boldsymbol{x}_i) - \nabla f_{\hat{k}(\boldsymbol{x}_0)}(\boldsymbol{x}_i)$
       $f_k' \leftarrow f_k(\boldsymbol{x}_i) - f_{\hat{k}(\boldsymbol{x}_0)}(\boldsymbol{x}_i)$
5    **end**
6    $\hat{l} \leftarrow \arg\min_{k \neq \hat{k}(\boldsymbol{x}_0)} \frac{|f_k'|}{\|\boldsymbol{w}_k'\|_2}$
     $\widetilde{\boldsymbol{r}} \leftarrow \frac{|f_{\hat{l}}'|}{\|\boldsymbol{w}_{\hat{l}}'\|_2^2} \boldsymbol{w}_{\hat{l}}' \ \widetilde{\boldsymbol{x}}_i = \boldsymbol{x}_i + \widetilde{\boldsymbol{r}}$
     $\boldsymbol{w}_i \leftarrow \nabla f_{k(\widetilde{\boldsymbol{x}}_i)}(\widetilde{\boldsymbol{x}}_i) - \nabla f_{k(\boldsymbol{x}_0)}(\widetilde{\boldsymbol{x}}_i)$
     $\boldsymbol{x} \leftarrow \boldsymbol{x}_0 + \frac{(\widetilde{\boldsymbol{x}}_i - \boldsymbol{x}_0)^\top \boldsymbol{w}_i}{\|\boldsymbol{w}_i\|^2} \boldsymbol{w}_i$
     $i \leftarrow i + 1$
7  **end**
8  **return** $\boldsymbol{r} = \boldsymbol{x}_i - \boldsymbol{x}_0$

---

**Algorithm 4:** SDF (1,3)

**Input:** image $\boldsymbol{x}$, classifier $f$.
**Output:** perturbation $\boldsymbol{r}$

1  Initialize: $\boldsymbol{x}_0 \leftarrow \boldsymbol{x}, \ i \leftarrow 0$
2  **while** $\hat{k}(\boldsymbol{x}_i) = \hat{k}(\boldsymbol{x}_0)$ **do**
3    **for** $k \neq \hat{k}(\boldsymbol{x}_0)$ **do**
4      $\boldsymbol{w}_k' \leftarrow \nabla f_k(\boldsymbol{x}_i) - \nabla f_{\hat{k}(\boldsymbol{x}_0)}(\boldsymbol{x}_i)$
       $f_k' \leftarrow f_k(\boldsymbol{x}_i) - f_{\hat{k}(\boldsymbol{x}_0)}(\boldsymbol{x}_i)$
5    **end**
6    $\hat{l} \leftarrow \arg\min_{k \neq \hat{k}(\boldsymbol{x}_0)} \frac{|f_k'|}{\|\boldsymbol{w}_k'\|_2}$
     $\widetilde{\boldsymbol{r}} \leftarrow \frac{|f_{\hat{l}}'|}{\|\boldsymbol{w}_{\hat{l}}'\|_2^2} \boldsymbol{w}_{\hat{l}}' \ \widetilde{\boldsymbol{x}}_i = \boldsymbol{x}_i + \widetilde{\boldsymbol{r}}$
7    **for** 3 *steps* **do**
8      $\boldsymbol{w}_i \leftarrow$
       $\nabla f_{k(\widetilde{\boldsymbol{x}}_i)}(\widetilde{\boldsymbol{x}}_i) - \nabla f_{k(\boldsymbol{x}_0)}(\widetilde{\boldsymbol{x}}_i)$
       $\boldsymbol{x}_i \leftarrow \boldsymbol{x}_0 + \frac{(\widetilde{\boldsymbol{x}}_i - \boldsymbol{x}_0)^\top \boldsymbol{w}_i}{\|\boldsymbol{w}_i\|^2} \boldsymbol{w}_i$
9    **end**
10   $i \leftarrow i + 1$
11 **end**
12 **return** $\boldsymbol{r} = \boldsymbol{x}_i - \boldsymbol{x}_0$

---

