# OpenReview forum: "SuperDeepFool: a new fast and accurate minimal adversarial attack"
_NeurIPS.cc/2024/Conference — NeurIPS 2024 poster_

### Official Review · Reviewer_Vkrz · 2024-07-09

**Soundness:** 4
**Presentation:** 4
**Contribution:** 3
**Rating:** 6
**Confidence:** 4

**Summary:**

The paper introduces SuperDeepFool (SDF), a new family of adversarial attacks aimed at testing the robustness of deep neural networks against minimal L2 perturbations. The proposed attacks generalize the DeepFool (DF) attack, improving both computational efficiency and effectiveness. The authors show that SDF surpasses existing methods in both effectiveness and efficiency, making it ideal for evaluating large models and improving adversarial training to achieve state-of-the-art robustness.

**Strengths:**

1. The paper introduces a novel method that leverages the geometric properties of minimal L2 adversarial perturbations, offering an innovative perspective on adversarial attacks.
2. The proposed method outperforms current state-of-the-art attacks by identifying smaller perturbations more efficiently, demonstrating superior effectiveness and computational efficiency.
3. The authors conduct solid and comprehensive experimental validation, illustrating the superiority of SDF across various scenarios and benchmarks.
4. The proposed method enhances the robustness of image classifiers through adversarial training, highlighting its practical applicability.

**Weaknesses:**

1. The paper focuses exclusively on L2 perturbations, which may not be the most practical or relevant threat model in real-world scenarios.
2. The proposed method builds upon DeepFool, an existing state-of-the-art white-box attack, which may limit the perceived novelty and contribution of the work.

**Questions:**

1. Can the insights gained from the proposed approach be extended to develop efficient black-box attacks?
2. Are there realistic and practical scenarios where the proposed attack can effectively evaluate the robustness of neural networks?

**Limitations:**

1. The paper focuses on a white-box threat model, which is less practical and realistic than the more challenging black-box threat model.
2. The paper focuses on L2 norm attacks, which may not adequately assess network robustness against realistic perturbations such as physical attacks.

---

> ### Author Rebuttal · Authors · 2024-08-06
>
> ## $\textbf{General comment:}$
> We sincerely thank the reviewer for their insightful and comprehensive assessment. We are particularly pleased that the reviewer recognized SuperDeepFool's core strengths: its computational efficiency, rigorous theoretical foundation, and robust empirical results demonstrated across a diverse range of datasets and tasks.
> ### $\textbf{Focus on $\ell_{2}$ perturbations}.$
> Given the 6,000-character constraint for this response and the substantial overlap between this section and the "The method is limited to ℓ2" section addressed for the $\texttt{"UfKL" reviewer}$, we kindly refer you to our response in that section. Should you require further clarification or elaboration, please do not hesitate to inform us.
> ### $\textbf{The proposed method builds upon DeepFool,  Novelty}$.
> Given the 6,000-character constraint for this response and the substantial overlap between this section and the **"Novelty"** section addressed for the $\texttt{"ojpB" reviewer}$, we kindly refer you to our response in that section. Should you require further clarification or elaboration, please do not hesitate to inform us.
> ### $\textbf{Develop efficient black-box attacks.}$
> We appreciate the reviewer's insightful question regarding black-box attacks. While SDF is primarily designed as a white-box attack, we believe the insights gained from our approach hold promise for developing efficient black-box techniques.
> As mentioned in our paper, the strategy employed by SDF approximating the decision boundary and finding minimal perturbations
>  shares similarities with the approach used in GeoDA [43], a black-box attack. This connection suggests that the **geometrical insights** underpinning SDF could potentially be adapted to the black-box setting.
> **However**, extending our method to black-box scenarios would **require** further research and modifications. Specifically, challenges such as estimating the decision boundary and its curvature **without** direct access to the model's gradients need to be addressed. We consider exploring these extensions as an exciting direction for future work and will investigate the feasibility of transferring our geometrically-inspired approach to black-box attacks.
> In the meantime, we believe the current contributions of SDF as a white-box attack are significant. By establishing a new benchmark for minimal-norm perturbations and demonstrating superior performance on robust models, SDF provides valuable insights into adversarial robustness that can inform the development of future defenses, regardless of the attack scenario.
>
> ### $\textbf{Realistic and practical scenarios}.$
> We appreciate the reviewer's emphasis on the importance of evaluating network robustness against realistic perturbations. While SuperDeepFool focuses on $\ell_{2}$ norm attacks, we believe it has significant practical relevance for evaluating and improving the robustness of neural networks in several scenarios:
>
> **Benchmarking and Comparing Defenses:** The $\ell_{2}$ norm provides a standardized and widely used metric for quantifying the magnitude of adversarial perturbations. SDF's ability to find minimal $\ell_{2}$ norm perturbations makes it a valuable tool for benchmarking different defenses and comparing their effectiveness in mitigating adversarial attacks.
>
> **Understanding Model Vulnerabilities:** Even though $\ell_{2}$ perturbations might not always directly translate to physical attacks, they can reveal underlying vulnerabilities in the model's decision boundaries. The **geometric insights** gained from SDF can help researchers identify regions of the input space where the model is particularly sensitive, guiding the development of more robust architectures.
>
> **Improving Adversarial Training:** Adversarial training is a common technique for enhancing robustness. By generating strong $\ell_{2}$ adversarial examples, SDF can be used to augment training data and make models more robust to a broader range of perturbations, including those that might not strictly adhere to the $\ell_{2}$ norm.
>
> **Bridging the Gap to Realistic Perturbations:** Research has shown that models robust to $\ell_{2}$ attacks often exhibit improved robustness to other types of perturbations as well, including some physical attacks. While the transferability is not perfect, the $\ell_{2}$ norm serves as a useful starting point for evaluating and improving robustness, as it **captures** the **general** concept of limiting the magnitude of perturbations [A, B].
>
> **Jailbreaking LLMs:**
> Jailbreaking Leading Safety-Aligned LLMs with Simple Adaptive Attacks [C]: This paper shows that even safety-aligned LLMs are vulnerable to simple adaptive attacks that exploit the model's internal representations. The attacks involve perturbing the prompts to induce undesirable behavior, which can be seen as a form of attack in the embedding space by utilizing ℓ2 distance.
>
> Furthermore, our work on SuperDeepFool contributes to the ongoing research effort to bridge the gap between theoretical ℓ2 attacks and realistic perturbations. The insights gained from understanding and mitigating ℓ2 attacks are valuable stepping stones toward developing defenses against more complex and diverse threat models.
> We acknowledge that ℓ2 norm attacks are not a comprehensive solution for evaluating robustness against all possible real-world scenarios. However, we believe that SuperDeepFool's contributions to the field, particularly its ability to find minimal ℓ2 perturbations and its superior performance on robust models, make it a valuable tool for both researchers and practitioners working on improving the security and reliability of neural networks.
>
> [A]: Tramèr,  The space of transferable adversarial examples. arXiv preprint.
>
> [B]: Moosavi-Dezfooli, Universal adversarial perturbations. CVPR 2017
>
> [C]: Andriushchenko, M., Jailbreaking leading safety-aligned llms with simple adaptive attacks. ICML 2024 Workshop on the Next Generation of AI Safety

---

> > ### Comment · Reviewer_Vkrz · 2024-08-12
> >
> > Thank you for your explanations and clarifications. I would like to maintain my score and recommend the paper for acceptance.

---

### Official Review · Reviewer_ojpB · 2024-07-10

**Soundness:** 3
**Presentation:** 2
**Contribution:** 2
**Rating:** 6
**Confidence:** 4

**Summary:**

The paper introduces SuperDeepFool (SDF), a new adversarial attack algorithm designed to evaluate the robustness of deep neural networks against L2-norm adversarial attacks. SDF generalizes the DeepFool (DF) attack by incorporating a projection step to find smaller perturbations while maintaining computational efficiency. The authors demonstrate that SDF outperforms many L2-norm attacks in terms of both effectiveness and computational cost on MNIST, CIFAR10 and ImageNet.

**Strengths:**

Strengths:

1.	Efficiency and theory justification: Based on the experimental results, SDF achieves a better trade-off between computational cost and attack efficiency compared to other baselines. Theoretical analysis is provided to support that such algorithm can converge to a point on the boundary.
2.	Experiments: The paper provides experimental results across three datasets, including a large one (ImageNet). It also shows that it can be combined with AutoAttack to further improve its strength.
3.	Robustness Improvement: The paper demonstrates that adversarial training using SDF-generated examples can better enhance the robustness image classifiers compared to DDN.

**Weaknesses:**

Weaknesses:

1.	Clarity Issues: Generally speaking, it is well-organized, but there are some minor things make it not very clear. For example, on Line 123, the sentence “on about the geometrical…” misses the beginning part. In Section 3, $f$ represents a binary classifier when explaining the theory and algorithms in 3.1, but it was defined as a C-class classifier in Section 2. It is better to distinguish the two using different notations.
2.	Novelty: In fact, leveraging the geometry of decision boundary to improve the efficiency of adversarial attacks is not a new idea. This has been explored in many blackbox attacks, such as [1], [2]. A discussion of the relationship between the proposed idea and previous works would help improve the work.

[1] Cheng, M., Singh, S., Chen, P., Chen, P. Y., Liu, S., & Hsieh, C. J. “Sign-opt: A query-efficient hard-label adversarial attack.” ICLR. 2020.

[2] Chen, Jinghui, and Quanquan Gu. "Rays: A ray searching method for hard-label adversarial attack." KDD. 2020.

**Questions:**

N/A

**Limitations:**

A discussion of limitation is not found in the paper.

---

> ### Author Rebuttal · Authors · 2024-08-06
>
> ## $\textbf{General comment:}$
> We are grateful to the reviewer for recognizing SuperDeepFool's efficiency, theoretical justification, and strong empirical performance across various datasets and tasks. We also appreciate the acknowledgment of its potential to improve adversarial robustness through training.
> ### $\textbf{Clarity Issues $\rightarrow$ notations}:$
> Thank you for your valuable feedback. We will address the clarity issues you mentioned in the final version. Please note that the main paper already includes pseudo-code algorithms in the appendix for all versions.
> ### $\textbf{Novelty:}$
> Thank you for your insightful suggestion. In $\texttt{lines 210-215}$ of the $\texttt{main manuscript}$, we have acknowledged that the method of iteratively approximating the decision boundary using a hyperplane, followed by the analytical determination of the minimal adversarial perturbation for a linear classifier, bears a resemblance to the approach employed by GeoDa $\textcolor{blue}{[43]}$ in black-box settings.
>
> SuperDeepFool is indeed inspired by DeepFool's core principles to **iteratively approximate the decision boundary** with a *hyperplane* and then analytically calculate the minimal adversarial perturbation for a linear classifier for which this hyperplane is the decision boundary. It is crucial to recognize the substantial advancements it introduces.
> SuperDeepFool reimagines the approach to adversarial perturbations through several **key innovations**:
>
> **Balancing Geometry and Optimization**:  A key insight of SuperDeepFool is *striking* a balance between the **geometrical inspiration** of deep neural networks and **modern optimization techniques**. We avoid the *pitfalls* of relying **solely** on **hyperparameter tuning** or **excessive iterations**. Instead, we leverage **both** geometrical understanding and efficient optimization to achieve **near-optimal** solutions with **fewer iterations**, which is crucial for the development of robust LLMs, for instance, and large models. These innovations result in a qualitatively different attack, one that **not only** outperforms DeepFool **but** also the entire landscape of current state-of-the-art attacks, as evidenced by our extensive empirical results. We believe this shift towards a more balanced approach, emphasizing **simplicity** and **geometrical understanding**, is crucial for the future of adversarial robustness.
>
> #### $\textbf{Key Insight:}$
> Broadly speaking, the approaches that employ **geometrical characterization** of deep neural networks can be categorized into white-box and black-box settings:
> 1. White-box Settings:
>     - For $\ell_{1}$ and $\ell_{0}$ norms, SparseFool [1] iteratively approximates the decision boundary with a hyperplane while controlling the level of **sparsity**. It employs adaptive coordinate-wise control ($\texttt{Qsolver}$). However, SparseFool encounters **issues** when dealing with **box constraints** in the range [0,1] (**clipping**) (𝜎-zero$\textcolor{blue}{[3]}$, APGD-$\ell_{1}$$\textcolor{blue}{[4]}$).
>
> 2. Black-box Settings:
>     - GeoDa [43] and qFool [2] aim to iteratively approximate the classifier's gradient with minimal query usage.
>
> The most significant contribution of SuperDeepFool is its ability to strike a balance between two critical characterizations of optimal adversarial perturbations: lying on the decision boundary and maintaining orthogonality.
>
> ### $\textbf{A discussion of limitation is not found in the paper.}$
> Our limitations are clearly reported in the paper. In particular, we had a section in the Appendix (N). Following the reviewer’s suggestion, we will emphasize those points further in the revised version of our work.
>
> ## References:
> [1]: Modas, A., Moosavi-Dezfooli, S., and Frossard, P. Sparsefool: a few pixels make a big difference. In CVPR, 2019.
>
> [2]: Liu, Y., Moosavi-Dezfooli, S. M., & Frossard, P. (2019). A geometry-inspired decision-based attack. In Proceedings of the IEEE/CVF International Conference on Computer Vision (pp. 4890-4898).
>
> [3]: Cinà, A. E., Villani, F., Pintor, M., Schönherr, L., Biggio, B., & Pelillo, M. (2024). 𝜎-zero: Gradient-based Optimization of ℓ0-norm Adversarial Examples. arXiv preprint arXiv:2402.01879.
>
> [4]: Croce, F., & Hein, M. (2021, July). Mind the box: ℓ1-APGD for sparse adversarial attacks on image classifiers. In International Conference on Machine Learning (pp. 2201-2211). PMLR.
>
> [43]: Ali Rahmati, Seyed-Mohsen Moosavi-Dezfooli, Pascal Frossard, and Huaiyu Dai. Geoda: a geometric framework for black-box adversarial attacks. In Proceedings of the IEEE/CVF conference on computer vision and pattern recognition, pages 8446–8455, 2020.

---

### Official Review · Reviewer_e1KW · 2024-07-11

**Soundness:** 3
**Presentation:** 4
**Contribution:** 3
**Rating:** 6
**Confidence:** 3

**Summary:**

The paper presents a novel, parameter-free and computationally efficient minimal-$L_2$ adversarial attack. Building on the DeepFool attack and incorporating a novel geometric perspective, the SuperDeepFool attack achieves state-of-the-art success rates on selected MNIST, CIFAR10, and ImageNet classifiers. The authors demonstrate that SuperDeepFool maintains computational efficiency while achieving higher success rates in fooling neural networks compared to similar attacks. Adversarial training experiments show that this approach improves the evaluation of neural network robustness against adversarial attacks.

**Strengths:**

The paper is clear, well-organized, and provides excellent geometric intuition and presentation. The illustrations are particularly effective in conveying complex concepts. The novel SuperDeepFool method demonstrates significant improvements over existing techniques by identifying the optimal adversarial point orthogonal to the decision boundary and employing an alternating optimization strategy. The paper's approach of combining the DeepFool attack with orthogonality optimization leads to higher success rates while maintaining computational efficiency. Additionally, the comprehensive comparisons and experiments highlight the effectiveness of SuperDeepFool across different models and datasets.

**Weaknesses:**

The paper has several areas that need further elaboration. The measurements to test optimality are simple and strightfoward, particularly regarding whether a $\gamma$ < 1 factor makes the adversarial perturbation fail. Hence, it is not clear whether a simple gamma optimization with DF could outperform SuperDeepFool (SDF). Comparisons in tables (e.g., Tables 2 and 3) lack clarity on success rates and how averages are computed. The comparison with Adversarial Training (AT) is very interesting but too brief, raising questions about the chosen perturbation size and the network's optimization against minimum-norm attacks. Additionally, the discussion on the comparison with auto attack is also too brief, and the rationale for switching from multitarget APGD (or other attacks) to single target SDF without evaluating the multitarget approach is unclear.

**Questions:**

1. Optimality Measurements: I believe comparing DF with simple $\gamma$ optimization against SuperDeepFool (SDF) is essential to start with, since it's much faster and is presented as a measure of optimality.
2. Comparison Tables: In tables such as 2 and 3, it is not clear if both DF and SDF achieve 100% success rates. Could you clarify the success rates and explain how averages are computed?
How do you account for variations across different models when presenting average success rates?
3. Adversarial Training (AT) Comparison: The comparison in the Adversarial Training section is quite brief. Could you elaborate on whether the evaluated network is state-of-the-art in terms of robustness and how it is optimized to defend against minimum-norm attacks?
The perturbation size of 2.6 chosen for SDF seems arbitrary. Could you explain the rationale behind this choice, especially given the convention of using 0.5-norm attacks (other AT networks don't claim to be robust to larger perturbations)?
4. Auto Attack Comparison: The discussion on the comparison with auto attack is also brief. Why did you choose to switch from multitarget APGD to single target SDF without evaluating the multitarget approach?
Does multitarget SDF achieve better robust accuracy?

**Limitations:**

The authors discussed the limitations of their experimental work - the limited evaluation of the SDF methods on a big variance of robust and non-robust classifiers, and extensions to targeted attacks and different norms. (no justifications for limitations on the checklist)

---

> ### Author Rebuttal · Authors · 2024-08-06
>
> ## $\textbf{General comment:}$
> We really appreciate the reviewer’s enthusiasm and acknowledgment of the significance of our work. We address the reviewer's concerns below.
> ### $\textbf{Optimality Measurements:}$
> Firstly, it is important to note that while avoiding overly perturbed perturbation is a **critical** property of minimal adversarial perturbation, **orthogonality** is another equally crucial aspect. In Figure 3 (right) of the main paper, we measured orthogonality by **bringing all perturbations** to the decision boundary using a **line search** at the **end** of the DF algorithm (Indeed, we optimize $\gamma$). This process is carried out **outside** the **main loop** of DF to ensure that the algorithm's **fooling rate** is **preserved**, as detailed in *lines 142 and 143 of the main paper*.
> Figure 3 (right) demonstrates that **even though** DF perturbations **lie on** the decision boundary and are **not** overly perturbed due to the line search, they still do **not** achieve optimal orthogonality. This raises a pertinent question: *How can we establish a connection between orthogonality and the minimality of perturbations?* We address this question in **two** ways:
> Firstly, by adding a line search to the end of DF, we optimize and compare the results of $\texttt{DF+line search}$ and SDF (*Results are presented in the attached rebuttal pdf. While line search in DF reduces perturbation norm, it doesn't guarantee optimal results, even with over 20 iterations.*). Secondly, we show that the **goal** of achieving minimal adversarial perturbation **extends beyond** minimality in terms of *median, mean, and quantity*; it also encompasses the **direction** of minimal adversarial perturbation:
> Adversarial training for robust models requires finding adversarial perturbations with **optimal direction**, not just optimal size. CURE [36] shows that adversarial training primarily regularizes curvature, prioritizing curvature-reducing directions. Our analysis shows that our model trained with optimal direction perturbations has considerably lower curvature, confirming the importance of optimal direction for model robustness.
> ### $\textbf{Comparison Tables:}$
> To compute Fooling Rates (FR), we follow the standard methodology in the literature by counting the number of samples that are *fooled* or *not fooled*. Specifically, a sample is considered fooled by adversarial perturbation if the model misclassifies it. Conversely, if the model correctly classifies a sample, the perturbation is deemed ineffective in fooling the classifier. It is important to note that before evaluating the algorithms, we **exclude** any samples that the model *initially* misclassifies.
> It is important to note that in Table 2, we do **not** report the FR or ASR. Instead, we present the results of the *level of orthogonality* obtained by DF and SDF. These results are identical to those shown in Figure 3 (Right) and Figure 4.
> ### $\textbf{Elaborate on evaluated networks}$:
> It is important to mention that the procedure of *"Vanilla Adversarial Training"* is explained in Appendix O of the main paper (lines 864 to 875).
> #### **Why is an adversarially trained (AT) model with strong minimum-norm attacks like SDF robust to others?**
> AT models with optimal perturbations can **potentially reduce** the model's curvature *concerning adversarial directions*. Specifically, when a model is made robust using optimal directions generated by **strong** minimum-norm attacks such as SDF, it is likely to exhibit robustness against perturbations produced by *sub-optimal* minimum-norm attacks, including DF, FMN, FAB, and others.
> The results are displayed in Table (6) of the main paper.
> ### $\textbf{The perturbation size of 2.6}.$
> The maximum perturbation budget ($\varepsilon$) of 2.6 is based on the standard ℓ∞ epsilon of 8/255, which translates to an ℓ2 norm of approximately 1.75. This value was slightly increased to allow for a wider range of perturbations, but as Table 4 shows, the impact of this choice is minimal due to the low median perturbation size.
> ### $\textbf{AA Comparison:}$
> It is important to note that APGD is not considered a minimum-norm attack but instead considered a norm-bounded attack.
> The main procedure of AA involves utilizing strong bounded-norm attacks such as APGD to find adversarial perturbations. Subsequently, a minimum-norm attack like FAB is employed to minimize the discovered perturbations accordingly. When utilizing APGD in the context of a minimum-norm version, it is advisable to incorporate a binary search [48]. To ensure 100% ASR, a sufficient budget is allocated for the binary search. The budget size depends on the dataset being used. By allowing the attacks to achieve a 100% ASR and conducting multiple binary search steps, a precision of 0.01 for the ℓ₂-norm can be achieved, as previously demonstrated by [48]. This will significantly raise the cost of computations. One might wonder why we choose to replace SDF with APGD instead of FAB as a minimum-norm attack in a set of AA. While our paper primarily focuses on other aspects, we aim to improve the time efficiency of the AA. Undoubtedly, the primary limitation of AA is its computational time.
> In general, combining a series of attacks and attempting to navigate through a sample can be accomplished using various different attacks (selecting the most powerful attack for each norm). However, a particularly intriguing scenario is one that can achieve this integration in the most efficient manner possible. So, we swap SDF for the critical point of a bottleneck for AA, known as APGD (line 733 of the appendix in the main paper). Nevertheless, this critique can still be directed towards our idea: How can we ensure the previous performance of AA with this modification? It is generally not possible to guarantee the performance of AA++ for all models. However, we can evaluate our notion through experimental evaluation, as we have done in our study.

---

> > ### Comment · Reviewer_e1KW · 2024-08-12
> >
> > I thank the authors for the interesting discussion. I hope the novelty of the orthogonality to the decision boundary will yield further interesting research.

---

### Official Review · Reviewer_UfKL · 2024-07-12

**Soundness:** 4
**Presentation:** 4
**Contribution:** 4
**Rating:** 7
**Confidence:** 4

**Summary:**

The paper introduces a new family of adversarial attacks called SuperDeepFool (SDF) attack, extending the well-known DeepFool (DF) attack. This novel approach strikes an proper balance between effectiveness and efficiency, and consistently outperforms existing methods in terms of both. Additionally, the method can be adapted for robustness evaluation and adversarial training.

**Strengths:**

1. The concept of integrating DF with minimal adversarial perturbations is both novel and intriguing.
2. The theoretical analysis is sound: DF iterations converge to a point on the decision boundary, while SDF iterations converge to a point orthogonal to the decision boundary.
3. Comprehensive experiments demonstrate SDF's remarkable performance compared to existing methods, in terms of both Fooling Rate (FR) and the number of gradient computations (Grads).
4. The experiments consistently show improvements over DF and minimum-norm attacks, together with its potential for adaptation in adversarial training (AT).

**Weaknesses:**

1. The method is limited to the $\ell_2$-norm.
2. Most of the experiments are conducted with $\varepsilon=0.5$.

**Questions:**

1. Can the method be generalized to the $\ell_p$-norm?
2. Do we expect similar performance for smaller $\varepsilon$, e.g. $8/255$?
3. Is the case SDF(m,$\infty$) also interesting?
4. How do we determine the number of iterations in SDF beforehand?

**Limitations:**

The potential limitations are outlined in the weaknesses and questions sections.

---

> ### Author Rebuttal · Authors · 2024-08-06
>
> ## $\textbf{General comment}$:
> We sincerely thank the reviewer for his/her positive assessment of our work, recognizing the novelty of our approach, the soundness of our theoretical analysis, and the strength of our empirical results. We are also pleased that the potential of SuperDeepFool for adversarial training is recognized.
> ### $\textbf{The method is limited to $\ell_{2}$ norm}$.
> First, we revisit the role of $\ell_{2}$ adversarial robustness in the robustness community. Then, we bring our results for the $\ell_{\infty}$ norm of SDF.
> As we discussed in $\texttt{lines 91 to 102}$ of the $\texttt{main text}$ of the paper, the reasons for using $\ell_{2}$ norm perturbations are **manifold**. We acknowledge that $\ell_{2}$ threat model may not seem remarkably realistic in practical scenarios (at least for images); however, it can be perceived as a basic threat model amenable to both theoretical and empirical analyses, potentially leading insights in tackling adversarial robustness in more complex settings. The fact that, despite considerable advancements in AI/ML, we are yet to solve adversarial vulnerability motivates part of our community to return to the basics and work towards finding fundamental solutions to this issue $\textcolor{blue}{[8, 24, 33]}$. In particular, thanks to their intuitive geometric interpretation, $\ell_{2}$ perturbations provide valuable insights into the geometry of classifiers. They can serve as an effective tool in the "interpretation/explanation" toolbox, a versatile role of our method, to shed light on what/how these models learn.
> It should be noted that the focus of our paper, as stated in the abstract, is on minimal $\ell_{2}$ norm perturbations.
> Nevertheless, we tried a $\ell_{\infty}$ version of SDF by replacing the $\textit{orthogonal projection}$ with the $\ell_{\infty}$ projection (Holder's inequality). The table in $\texttt{attached rebuttal pdf}$ shows our results for $\ell_{\infty}$ on M1 $\textcolor{blue}{[32]}$ and M2 $\textcolor{blue}{[47]}$. Our results show that this version of SDF also outperforms other algorithms in finding smaller $\ell_{\infty}$ perturbations (**the result is presented in the attached rebuttal pdf**). We will add this result to the Appendix.
> ### $\textbf{Do we expect similar performance for smaller $\varepsilon$, e.g. 8/255?}$ (Most of the experiments are conducted with $\varepsilon=0.5$).
> First, we should note the reason that we use $\varepsilon = 0.5$ for comparison between $\ell_{2}$ versions of AA and AA++ is that this budget ($\varepsilon = 0.5$) is a **standard** case for comparison models in robustbench $\textcolor{blue}{[12]}$ for CIFAR-10. Indeed, the primary models that robustbench $\textcolor{blue}{[12]}$ want to evaluate primary adversarially trained with that specific budget ($\varepsilon = 0.5$), so the **standard** way to evaluate their robustness is using that specific budget.
> For instance, the standard budget for $\ell_{\infty}$ on CIFAR-10 is $8/255$, and for $\ell_{2}$ on ImageNet, it is $\varepsilon = 3.0$. But in any case, we are grateful for your suggestion and will present our results for other budgets.
> #### **$\varepsilon = 0.3$ for $\ell_{2}$ on CIFAR-10 result is presented in attached rebuttal pdf.**
> #### **Comparison between our AT model and $\textcolor{blue}{[47]}$ beyond other $\varepsilon$ :**
> We presented these results in $\texttt{line 363}$ of the main text and **Tables 12** and **13** of the $\texttt{appendix}$ for both the $\ell_{\infty}$ and $\ell_{2}$ versions of AA across a wide range of $\varepsilon$ values.
> ### $\textbf{How do we determine the number of iterations in SDF beforehand?}$
> It is important to note that SDF is a **parameter-free** attack. This implies that we do not set a fixed number of iterations for SDF; instead, the algorithm runs **until** it identifies the **adversarial point**. This factor significantly contributes to the exceptionally high speed of SDF. The results presented in the tables demonstrate that SDF computes a substantially **smaller** number of gradients compared to other algorithms to identify the adversarial point. To ensure a **fair** comparison between SDF and other fixed iteration attacks, we standardized the maximum number of iterations for the algorithm. Specifically, when comparing **fixed iteration** algorithms that operate with $100$ iterations, we set the maximum number of iterations for SDF to $100$ and vice versa.
> Although the **iteration-free** property of an algorithm can enhance its convergence speed, a **critical** question arises: $\textit{“When the algorithm terminates upon finding the adversarial point, how can we \textbf{ensure} that this adversarial point is optimal?”}$ This is one of the key **criticisms** that can be raised against **DeepFool**. As demonstrated in $\texttt{Figure 3 (Left)}$ of the main paper, DeepFool's perturbations are **not** optimal and tend to be overly perturbed (extrapolated).
> To address this issue, we can employ a **line search** at the end of the DeepFool algorithm to ensure that the perturbations are **not** excessively large and remain on the decision boundary. However, as shown in $\texttt{Figure~3(Right)}$ of the main paper, line search $\textit{alone}$ **cannot** resolve the issue of **orthogonality**.
>
> ### $\textbf{Is the case SDF(m, $\infty$) also interesting?}$
> We appreciate your insightful perspective. However, we must clarify that this cannot be considered an interesting case. Let us revisit the rationale behind the SDF$(m, n)$ and explain why, although SDF$(\infty, 1)$ is a case of interest and controversy, SDF$(m, \infty)$ **cannot** be considered interesting. The **orthogonal projection** step in SDF$(\infty, 1)$ crucially **reduces** perturbation sizes, ensuring minimal adversarial perturbations. By iteratively using this projection, the optimal direction is **preserved**, leading to minimal perturbations that **cannot fool** the classifier, thereby maintaining classifier accuracy.

---

> > ### Comment · Reviewer_UfKL · 2024-08-08
> > **Response to the rebuttal**
> >
> > Thank you to the authors for their thorough explanations and clarifications. The rebuttal effectively addressed my concerns. As a result, I would like to maintain my scores and recommend acceptance.

---

### Author Response · Authors · 2024-08-06
**General Comment**

We kindly thank all the reviewers for their time and for providing valuable feedback on our work. We appreciate that reviewers have pointed out that our work is novel and intriguing (Reviewer $\texttt{UfKL}$), clear, well-organized, and provides excellent geometric intuition and presentation (Reviewer $\texttt{e1KW}$), and with justified theoretical insights (Reviewer $\texttt{ojpB}$ ), and that our results are solid and impressive (Reviewer $\texttt{Vkrz}$).

We hope that these results and the clarifications detailed in the individual comments given to each reviewer will effectively address the concerns raised during the review process. We remain available to engage in any further discussions that may arise, and we thank you once again for your comments.

---

### Author Rebuttal · Authors · 2024-08-06

We attached a pdf file containing the desired results.

---

### Decision · Program_Chairs · 2024-09-25

**Decision:**

Accept (poster)

**Comment:**

**Summary of the paper**

This paper proposes improving a well-known adversarial example generation attack, DeepFool. Specifically, the authors observe that DeepFool does not generate an adversarial perturbation that is orthogonal to the decision boundary, suggesting that DeepFool is generating suboptimal adversarial examples. To address this issue, the authors propose a new method that projects the adversarial perturbation toward the optimal perturbation direction. The authors provide a few theoretical results and extensive experiments to support the improvement.

**Summary of the discussion**

* All the reviewers mentioned that this paper's strengths are in the novelty of the idea, the quality of the theoretical analysis, and the comprehensive experiments, which showed impressive performances.
* Reviewer UfKL points out two weaknesses.
  * First, the paper only focused on $L_2$ perturbations. The authors claim that the $L_2$ perturbation can be perceived as a basic threat model amenable to theoretical and empirical analyses. They also added the results for the $ L_\infty$ version of SuperDeepFool.
  * Second, most experiments are conducted on a fixed attack radius. Regarding the second weakness, the author pointed out that it is the standard setup used to evaluate robustness. The author also presented results for a different parameter setting in their revision.
  * After the discussion, the reviewer maintained the initial rating of 7.
* Reviewer e1KW points out three weaknesses.
  * First, the optimality measurement for the DeepFool is too simple, and the DeepFool algorithm might provide better perturbations than the proposed method. Regarding the optimality measurement, the authors emphasized that orthogonality is another crucial aspect that should be considered in the optimality definition, so adjusting the $\gamma$ parameter would not make the DeepFool perturbations stronger than the proposed method.
  * Second, the comparisons in some experimental results seem unclear. The authors clarified the tables that the reviewer found confusing.
  * Third, the comparisons to adversarial training and auto-attacks are too brief. The authors justified why their comparisons to adversarial training and auto-attacks are sufficient.
* Reviewer ojpB points out two weaknesses.
  * First, the reviewer points out some minor writing issues that should be improved. The authors accepted the writing improvements.
  * Second, the proposed idea used an idea explored by many existing methods. Regarding the originality of their work, the authors admitted that their idea is mostly based on DeepFool’s core principle. but they also clarified their key innovations in improving DeepFool’s idea.
  * After the discussion, the reviewer maintained the initial rating of 6.
* Reviewer Vkrz points out two weaknesses.
  * First, the paper only focused on $L_2$ perturbations.
  * Second, the proposed method merely improves DeepFool rather than an original attack algorithm.
  * Because Reviewer Vkrz mentioned the same weaknesses as Reviewer UfKL and Reviewer ojpB, the authors simply referred to their rebuttal arguments against those reviewers.
  * After the discussion, the reviewer maintained the initial rating of 6.

**Justification of my evaluation**

* My rating is 7.
* In general, the contribution seems enough for paper acceptance. This is a well-written paper.
* The authors communicated effectively during the discussion. However, the arguments were not enough to change any reviewer’s rating. In my opinion, the reviewer’s initial ratings were accurate, and I don’t think it would be possible to increase the rating further.
* I also agree with the reviewers that the proposed method is an improvement of a well-known method rather than a new invention of a new algorithm. Considering this lack of novelty, I’d not recommend this paper for the spotlight.

I’m confident about my evaluation and recommend accepting this work (poster).